# A deep learning-based stripe self-correction method for stitched microscopic images

Shu Wang [1,2,3,9], Xiaoxiang Liu[2,9], Yueying Li[1,9], Xinquan Sun[1], Qi Li[2], Yinhua She[2], Yixuan Xu[1], Xingxin Huang[3], Ruolan Lin[4], Deyong Kang[5], Xingfu Wang[6], Haohua Tu[7,8], Wenxi Liu [2]✉, Feng Huang [1]✉ & Jianxin Chen [3]✉

Stitched fluorescence microscope images inevitably exist in various types of stripes or artifacts caused by uncertain factors such as optical devices or specimens, which severely affects the image quality and downstream quantitative analysis. Here, we present a deep learning-based Stripe Self-Correction method, so-called SSCOR. Specifically, we propose a proximity sampling scheme and adversarial reciprocal self-training paradigm that enable SSCOR to utilize stripe-free patches sampled from the stitched microscope image itself to correct their adjacent stripe patches. Comparing to off-the-shelf approaches, SSCOR can not only adaptively correct non-uniform, oblique, and grid stripes, but also remove scanning, bubble, and out-of-focus artifacts, achieving the state-of-the-art performance across different imaging conditions and modalities. Moreover, SSCOR does not require any physical parameter estimation, patch-wise manual annotation, or raw stitched information in the correction process. This provides an intelligent prior-free image restoration solution for microscopists or even microscope companies, thus ensuring more precise biomedical applications for researchers.

Fluorescence microscope is an indispensable tool for biomedical research, which can be used to obtain auxiliary information with high spatial resolution, such as intracellular biological processes or histopathological features[1,2]. At present, the most common approach to obtain large-scale high-resolution microscopic images is to continuously stitch multiple tiles or field of views (FOVs)[3,4]. However, the stripes, shadings, and even artifacts often remain in the stitched fluorescence images, especially in the weak signal regions of label-free or large-scale stitched images[5–9]. According to our imaging experience and previous research results, it has been discovered that, even for the most stable commercial nonlinear optical

microscopic instruments, the diverse stripes may still exist in the acquired images[10]. Optical engineers often underestimate the impact of stripes on biomedical researches. A survey of 170 microscope users found that ignoring the stripe correction procedure tends to result in a 35% increase in false and missed detection of cells in images[11]. In addition, the images severely contaminated by stripes usually will be discarded by microscopists. The feature analysis on the image content around stripes is carefully avoided by researchers as well. Therefore, the stitched stripes not only reduce the image quality, but seriously introduce considerable bias to the downstream quantitative analysis.

[1]College of Mechanical Engineering and Automation, Fuzhou University, Fuzhou 350108, China. [2]College of Computer and Data Science, Fuzhou University, Fuzhou 350108, China. [3]Key Laboratory of OptoElectronic Science and Technology for Medicine of Ministry of Education, Fujian Provincial Key Laboratory of Photonics Technology, Fujian Normal University, Fuzhou 350007, China. [4]Department of Radiology, Fujian Medical University Union Hospital, Fuzhou 350001, China. [5]Department of Pathology, Fujian Medical University Union Hospital, Fuzhou 350001, China. [6]Department of Pathology, The First Affiliated Hospital of Fujian Medical University, Fuzhou 350005, China. [7]Beckman Institute for Advanced Science and Technology, University of Illinois at Urbana-Champaign, Urbana, IL 61801, USA. [8]Department of Electrical and Computer Engineering, University of Illinois at Urbana-Champaign, Urbana, IL 61801, USA. [9]These authors contributed equally: Shu Wang, Xiaoxiang Liu, Yueying Li. ✉e-mail: wenxiliu@fzu.edu.cn; huangf@fzu.edu.cn; chenjianxin@fjnu.edu.cn

Some optical approaches can reduce or even avoid the stripe effect during imaging process, such as optimizing the beam quality and excitation optical path, or adjusting the levelness of the stage. However, the configuration of the microscope system requires professional optical engineers to operate. Increasing the laser power excites the fluorescence signal to a greater extent, but high power will increase the risk of photodamage to specimen. Besides, commercial microscopic software can zoom out the scanning range of the scanner to only image the central shading-free FOV, or increase the stitching overlap of two adjacent tiles. Nevertheless, scaling the single tile will also reduce the size of the final stitched image. Although optical engineers have utilized Bessel beams and strip mosaicking to remove vertical stripes while increase scanning speed[12,13], the horizontal stripes still exist, and these methods may result in a decrease in spatial resolution. Thus, there is always a trade-off between FOV and resolution for optical correction methods. As a result, image post-processing approach is the preferred solution for stripe correction.

In general, the existing stripe correction algorithms can be divided into prospective approaches and retrospective approaches[11,14]. Prospective approaches require additional reference images to estimate the effective illumination variation of the target image during the image acquisition process. Some commercial fluorescence microscopes provide customized reference microslides to acquire reference images. In fact, imaging parameters vary with the specimen type, quality, and laser power for each experiment. Moreover, the effective imaging time of different samples is limited (e.g., live cells, immunofluorescence, and fresh tissue), so it is difficult to obtain accurate reference images under the same experimental conditions within the limited time. In contrast, retrospective approaches only utilize actual acquired images to construct the correction model. Two of the representative state-of-the-art algorithms are BaSiC[14] and bundled correction software from commercial microscope (e.g., Zen[15], Zeiss). They remove the stitched stripes by correcting the shading of each tile, which perform well for uniform stripe images when each stitched tile has the same shading pattern, that is, falloff of intensity from the center of the image. However, complex and multi-type stripes usually appear in practical imaging experiments. It is difficult to fit a generic model to correct each shading tile in non-uniform stripe images due to different shading patterns caused by a combination of multiple factors, such as misaligned optics, beam quality, and uneven sample or stage. Moreover, these methods require the tile size to be known or access raw stitching data to obtain the optimal corrected images, but such prior information is usually not intentionally recorded in experiments. Besides, once the images are stitched online, commercial correction software cannot perform correction operations even if the microscopist has recorded the stitched information. With the recent progress of deep learning technique, deep neural network-based models can also be applied for this task. Most relevant to our work, a fully convolutional network-based method[16] was proposed to process color microscopic image with uneven illumination. Yet, these types of methods are based on supervised learning scheme which demands raw stitching tiles and the well restored images processed by experts to form a large number of registered image pairs for training their models. The dependency on large amount of supervised training data will limit the use of the method in real scenarios.

In this paper, we propose a deep unsupervised Stripe Self-Correction method (SSCOR), which accomplishes the task of stripe correction and artifact removal based on the input stitched image itself through a self-correction procedure consisting of proximity sampling, adversarial self-training, and local-to-global correction schemes. In principle, unlike previous methods that aim to process image tiles, SSCOR is uniquely formulated as patch-based shading correction method, and it is able to adaptively handle the various patterns of stripes and even artifacts existed in the stitched fluorescence images. In contrast to previous deep learning-based shading correction methods, SSCOR can be applied in situations where there is insufficient training data with only one or a few given images. Besides, the self-correction process of our method does not require any optical mechanical structure adjustment or optical path design, while it does not rely on the estimated physics parameters or raw stitched information either. Comparing to off-the-shelf approaches, we demonstrated that SSCOR has achieved the state-of-the-art performance on four image datasets with different modalities, which provided an intelligent image quality optimization solution to facilitate more precise downstream biomedical applications for researchers.

## Results
### SSCOR workflow
As illustrated in Fig. 1a, the purpose of SSCOR is to restore the content of the stitched fluorescence images via adaptively removing various stripes and artifacts, including non-uniform, oblique, and grid stripes, as well as scanning, bubble, and out-of-focus artifacts. To accomplish this purpose, we performed validation experiments on images of different histopathological features and modalities, including label-free multiphoton microscopy (MPM) datasets, labeled fluorescence datasets[17], and stimulated Raman scattering (SRS) image datasets[18]. The workflow of SSCOR is shown in Fig. 1b. SSCOR follows the divide-and-conquer strategy, in which the stripe correction of the stitched images can be divided into the stripe correction task for all the sub-regions, i.e., image patches. Given any stitched image, a proposed proximity sampling strategy is applied on the first stage, in which a pair of adjacent normal and anomaly patches that are respectively sampled on and off the stripes or the regions with artifacts (Supplementary Fig. 1), in order to prepare sufficient approximately registered positive/negative training data for the stripe correction model. This patch sampling strategy demands little manual effort in the preprocessing stage comparing to those methods that rely on the raw information of image tiles, while it makes the proposed model tolerant towards the mildly imprecise or roughly estimated partition of tiles. On the second stage, an adversarial self-training scheme is employed to train the stripe correction model. In particular, a deep convolutional neural network, i.e., the stripe correction network, is established for restoring anomaly patches with stripes or artifacts. To preserve the quality of stripe correction, the restored patches gained by the stripe correction network are assessed by a discriminator network with the reference to the corresponding normal patches sampled in their proximity, since the patches in proximity usually contain similar textures and illumination condition. Moreover, SSCOR also introduces an auxiliary task, in which we reciprocally synthesize the restored image patches into anomaly ones by composing similar stripe pattern with them and evaluate the consistency between the original patches and their synthesized ones, which can effectively constrain the stability of stripe correction. To do so, another neural network, i.e., the stripe synthesis network, is applied to accomplish this purpose. Last, on the local-to-global correction stage, the whole-slide images are partitioned into multiple overlapping patches in the sliding-window manner and fed into the stripe correction network to obtain corrected patches. After that, all the corrected patches are merged to form the final corrected result.

### SSCOR adaptively corrects different types of stripes
Non-uniform stripes, grid stripes, and oblique stripes are commonly observed in stitched fluorescence images. As illustrated in Fig. 2, the correction results of SSCOR were compared against the off-the-shelf methods CIDRE[11] and BaSiC[14] as well as the software ZEN[15]. First of all, the label-free MPM image of breast cancer showed typical non-uniform stripes in Fig. 2a. Diverse shading patterns of each tile lead to non-uniformity, caused by uncontrollable factors such as the condition of specimen or instrument. The major challenges rest in two aspects: 1) strong signal area needs to be properly corrected, otherwise the

stripes will appear to be more obvious; 2) weak signal area also require to be enhanced, so as to make the entire image consistent. The magnified area showed that SSCOR performed significant optimal de-stripe effects than CIDRE[11], BaSiC[14], and ZEN[15]. Second, for the grid stripes formed by stitching uniform tile shadings, SSCOR also showed superior correction ability in the representative MPM images on liver cancer (arrows in Fig. 2a). As observed, the existing methods over-enhanced the signal on the stripes. Thanks to the proximity sampling scheme, SSCOR produced smoother results on the stripes than the others. Note that, without raw tile information as prior knowledge, ZEN cannot produce any results. Last, as the pathological tissues are often adhered on glass slide askew, the captured image may need to be rotated properly to meet a downstream application need, so that the stripes will be slightly oblique. This oblique stripe can be commonly

observed in high-throughput fluorescent labeled images of mouse brain datasets[17]. As observed, SSCOR also achieved the best correction effect on the oblique stripes, while other methods were almost ineffective for this stripe.

To quantify the correction quality, we used the intensity profiles along the region of interest (ROI), i.e., the area within white rectangle in the oblique stripes, are given in Fig. 2b. The intensity profiles of raw images show obvious fluctuations as it transitions from non-striped areas (white areas) to striped areas (grey areas). Compared with other methods, SSCOR was able to correct the intensity of fluctuations more effectively while preserving tissue characteristic information. For further verification, the inverse coefficient variation (ICV)[19,20] was adopted to evaluate the de-stripe ability of different correction algorithms. SSCOR achieved the best performance on these three types of stripe

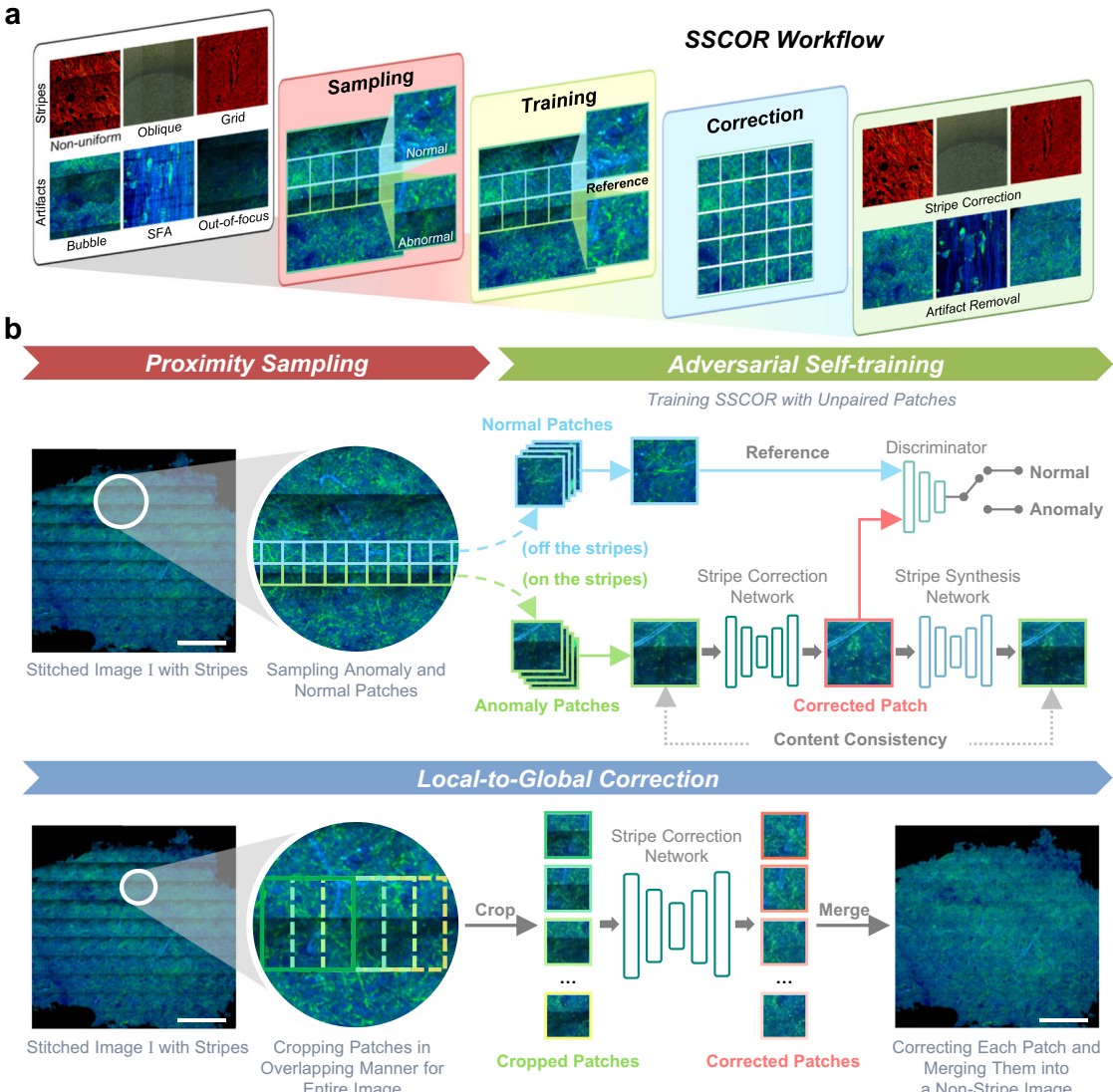

**Fig. 1 | Illustration of the proposed SSCOR workflow. a** Overview of SSCOR workflow. SSCOR is able to adaptively cope with various stripes (e.g., non-uniform, oblique, and grid stripes) and artifacts (e.g., bubble-like, out-of-focus, and scanning fringe artifacts) via the three stages of sampling, training, and correction. **b** The SSCOR framework consists of three stages to accomplish stripe self-correction, including proximity sampling, adversarial self-training, and local-to-global correction. First of all, given a stitched image with stripes, SSCOR samples adjacent normal and anomaly patches (in blue and green boxes) on and off the stripes for preparing training data, respectively. Next, SSCOR acquires the ability of stripe correction and artifact removal through patch-based adversarial self-training. In

overall, SSCOR is composed of sub-networks, i.e., stripe correction network, stripe synthesis network, and discriminator, and they play different roles on the training phase. Concretely, the stripe correction network restores anomaly patches and the stripe synthesis network synthesizes the corrected patches into anomaly ones, which are subject to consistency constraints so as to preserve the original image content. Additionally, with the normal patches as reference, the discriminator assesses the quality of corrected patches. Last, on the local-to-global correction stage, all the patches of the given stitched images will be corrected by the well-trained stripe correction network, and the results will be merged to form a stripe-free image as the final result. Scale bars: 1 mm.

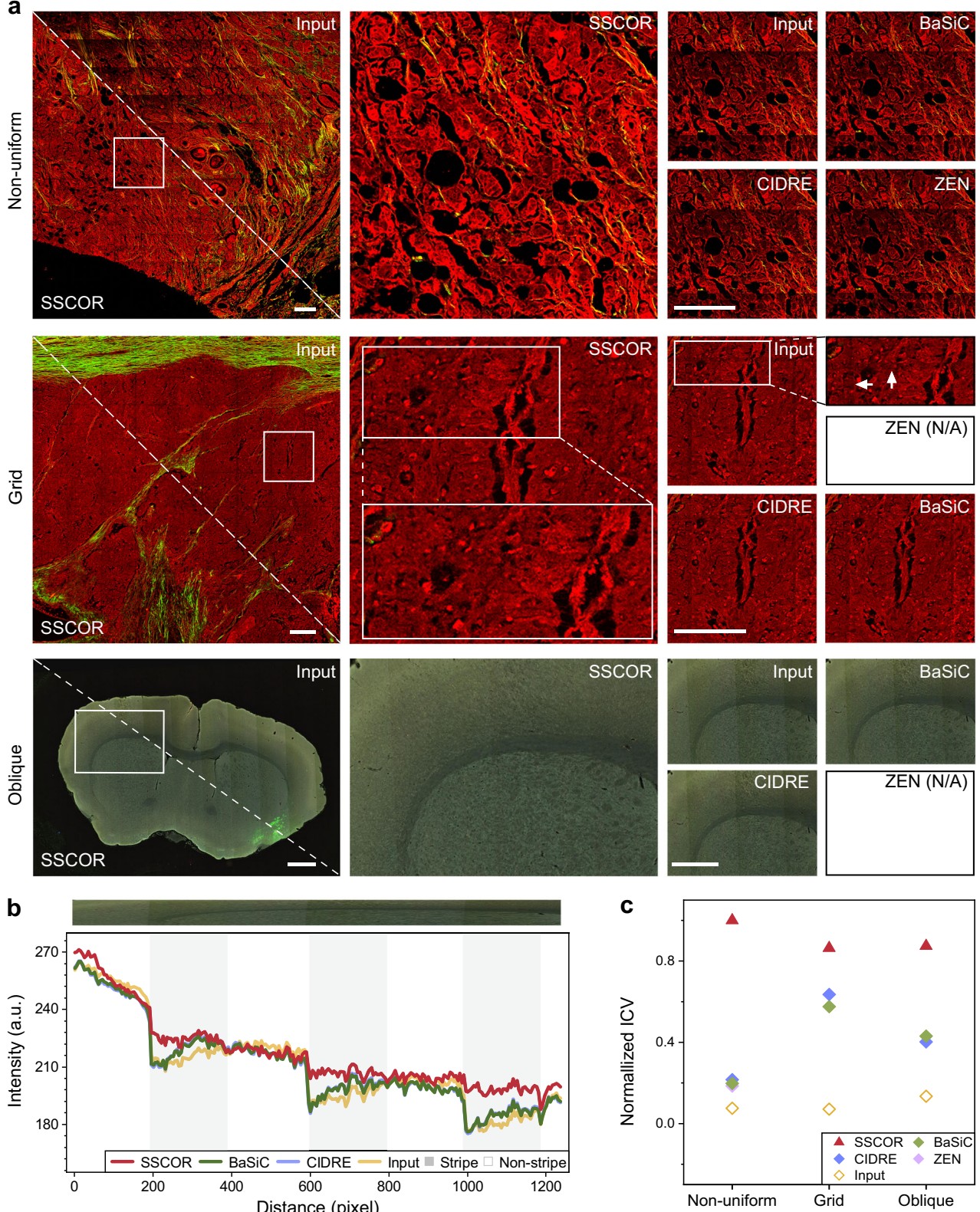

**b**

**c**

images (Fig. 2c). For more details, the intensity profiles of the non-uniform stripes and grid stripes, as well as the other typical stripe correction examples were shown in Supplementary Fig. 2. Besides, we illustrated the correction results across different intensity levels of heterogeneous images in Supplementary Fig. 3, which demonstrate that SSCOR does not overestimate or underestimate the fluorescence

signal in stripe regions while still preserving the intensity contrast of tissue components in non-stripe regions.

**High tolerance towards imprecise prior stitched information**
Practically, researchers often crop or rotate the stitched images to select desired ROIs, as illustrated in the schematic diagram from Fig. 3.

**Fig. 2 | The correction results of different types of stripes. a** Non-uniform, grid, and oblique stripe are the most common stripes in imaging experiments. Compared with the off-the-shelf method via zooming in the local details, SSCOR shows the ability of stripe correction on large-scale stitched multiphoton image of breast cancer (non-uniform stripe), multiphoton image of liver cancer (grid stripe), and fluorescence images of mouse brain (oblique stripe). White arrows indicate the positions of grid stripe. Notably, the Zeiss correction method is not applicable (N/A) because there are no raw files for the grid and oblique stripe images. Red and green pseudo-color respectively represent the two channels of multiphoton microscope. Scale bars of non-uniform stripe and grid stripe: 200 μm. Scale bars of oblique stripe: 1 mm. **b** The intensity profiles along the area within the white

rectangles in the oblique stripes demonstrate that SSCOR corrects the intensity fluctuations more effectively, while preserving tissue characteristic information. **c** The inverse coefficient variation (ICV) further quantitatively evaluates the de-stripe ability. A larger ICV corresponds to the flatter intensity profile and indicates a better correction quality. SSCOR outperforms the comparison methods on these three types of stripe images ($n_{\text{Non-uniform}} = 15$, $n_{\text{Grid}} = 14$, and $n_{\text{Oblique}} = 41$; $n$ represents the quantity of the stripe images. The size of the images varies depending on the stripe direction and the homogeneity of histopathological features). The markers represent the mean values of the normalized ICV. Source data are provided as a Source Data file.

Yet, there may exist errors in the process of manual cropping or extraction, due to unknown original tile size, while rotation may also increase the difficulty of image restoration and thus result in oblique stripes. Most existing methods are sensitive towards the imprecise prior stitched information, while the proposed SSCOR shows higher error tolerance. To demonstrate the ability, we manually cropped the MPM image of cerebral vascular malformation (uniform stripe image in Fig. 3a), and rotated the fluorescence image of mouse brain (oblique stripe in Fig. 3b) to simulate the procedure.

As observed, SSCOR showed higher tolerance and robustness against the stripes with cropping or rotation, according to the metrics of ICV under different cropping and rotation conditions. In contrast, using BaSiC, the stripes still remained in the correction images under these situations. Concretely, SSCOR can maintain stable correction as the cropping error is no more than 16% of the tile size, or the angle of the oblique stripes is less than 7 degrees. In addition, the intensity profiles showed that SSCOR can improve the signal intensity on the image regions with stripes while well preserving the image contrast. Apart from the corrected shading regions, SSCOR minimized the intensity variations in the non-stripe regions (Fig. 3c, d). Supplementary Fig. 4 displayed the comparison results of CIDRE, and detailed results under greater rotation conditions. Besides, SSCOR also showed tolerance towards the imprecise user-defined abnormal region for bubble-like artifacts (Supplementary Fig. 5). To sum up, SSCOR not only requires little raw information to correct stripes, but also demonstrates high tolerance and robustness against imprecise prior image adjustments.

## SSCOR removes special artifacts on stripes

Besides from the stripes that may existed in the stitched images, there are empirically three types of special artifacts on stripes during image acquisition, as shown in Fig. 4. The first is the out-of-focus artifact caused by the unevenness of the specimen, which significantly reduces the signal intensity of images. The second is the scanning fringe artifacts (SFA) produced by a high-speed galvo-resonant scanner imaging[21]. The third is the bubble-like artifact mainly caused by local tissue moisture loss, which is often observed in laser photodamage or the unsealed specimen. These artifacts deteriorate the image quality and make the histopathological features unusable for downstream applications. Therefore, we synthesized these special artifacts and stripes based on SRS images of human brain tumors[18] (Fig. 4a). The procedure of artifact synthesis was described in Supplementary Fig. 6 and Supplementary Note 1.

Unfortunately, most tile-based stripe correction methods like BaSiC, can hardly be applied to this task, since these artifacts are not caused by uniform optical patterns within tiles. For a thorough comparison, we also introduced latest deep learning-based image enhancement methods, including ZeroDCE[22], Neighbor2Neighbor[23] and Mask-ShadowGAN[24], to perform these tasks. The reasons and limitations of using these deep learning comparison methods were detailed in Supplementary Note 2. In Fig. 4b, SSCOR exhibited the ability to correct both the artifacts and stripes, which can restore weak tissue signals from the out-of-focus areas while reserving the color

fidelity of the original image. Besides, SSCOR can remove the SFA, and recover the partial original tissue signal from the bubble artifact. Our correction results restored most of the histopathological features that could be used for downstream analysis. Quantitatively, the intensity profiles within the out-of-focus magnified areas demonstrated the capability of SSCOR on artifact restoration and stripe correction (Fig. 4c). Moreover, through the calculation of peak signal-to-noise ratio (PSNR) and structural similarity index (SSIM) of the corrected image, SSCOR-corrected images have better image quality than other methods (Fig. 4d). We also showed the restoration results of real out-of-focus artifact, bubble artifact, and photobleaching artifact in the MPM images (Supplementary Fig. 7). In addition, to further evaluate the effectiveness of different correction methods, we conducted a user study involving 50 participants from diverse backgrounds (Supplementary Note 3). The results of the user study were presented in Supplementary Table 1 and 2, and Supplementary Fig. 8, which demonstrated that SSCOR significantly outperformed other methods for stripe correction and artifact removal in each discipline.

## SSCOR-corrected images benefit downstream tasks

Stripes and artifacts in the images have a negative impact on downstream application analysis. Therefore, we validated the benefits of SSCOR-corrected images for downstream tasks through four common post-processing applications (Fig. 5a), including virtual pathological staining, cell classification, automatic cell counting, and collagen signatures extraction.

First of all, virtual pathological staining of fluorescent images has been proven to promote the clinical development of microscopic imaging techniques[25-28]. Therefore, we utilized a promising deep-learned virtual staining model, UTOM[27], to transform non-uniform stripe MPM images of cerebral vascular malformations into H&E-stained images (Fig. 5b). However, the stripes still exist on the images transformed by the virtual staining algorithm owing to the uncorrected stripes in the stitched fluorescent images. More seriously, the stripes also cause false staining of tissue characteristics (i.e., the mis-generated cells as highlighted by the arrows in Fig. 5b), which may inadvertently lead to misdiagnosis by pathologists. By contrast, SSCOR-corrected images were able to suppress these effects brought by the stripes. Second, we corrected the stripes in real breast H&E images collected using a pathology slide scanner (Motic VM1000). In Fig. 5c, SSCOR displayed a comparable correction performance to BaSiC. We found that the slight difference in stripe correction would also affect the classification results. In specific, we conducted the cell classification for the uncorrected, BaSiC-corrected, and SSCOR-corrected images, respectively. As observed in the enlarged local image region, the SSCOR-corrected image showed the best classification results. We noticed that the lymphocytes highlighted by arrows were correctly classified in the SSCOR-corrected image (the cells with black contour), while BaSiC-corrected and uncorrected images result in the errors of misidentifying them as fibrocytes (the ones with yellow contour). In Supplementary Fig. 9a, we showed the complete corrected source image of Fig. 5c and highlighted the cell classification result of another stripe-free region. In addition, the classification

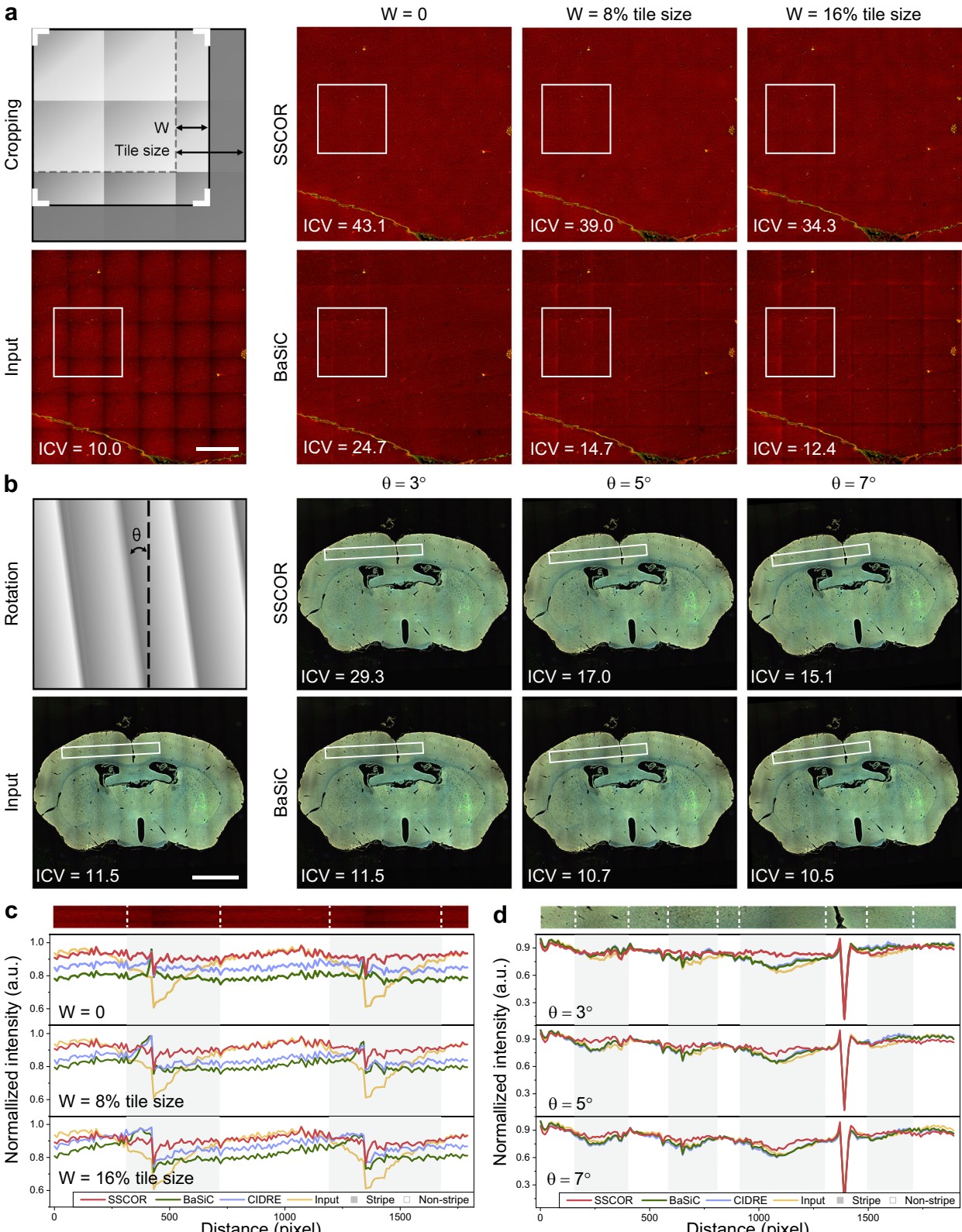

**Fig. 3 | SSCOR has high tolerance towards imprecise prior stitched information.** **a** For those applications that need cropping regions of interest (ROIs) from images, as illustrated in the schematic diagram, SSCOR maintains a more stable correction for typical uniform stripe images than BaSiC when the cropping error does not exceed 16% of the tile size. Scale bar: 1 mm. **b** Similarly, for the image rotation, SSCOR shows a more stable correction as the angle of the oblique stripes (θ) is less than 7 degrees. Scale bar: 2 mm. **c**, **d** The inverse coefficient variation (ICV), and the intensity profiles within the rectangular areas in **a** and **b** quantitatively demonstrate that SSCOR shows higher tolerance and robustness against the imprecise prior image adjustments than other methods. Source data are provided as a Source Data file.

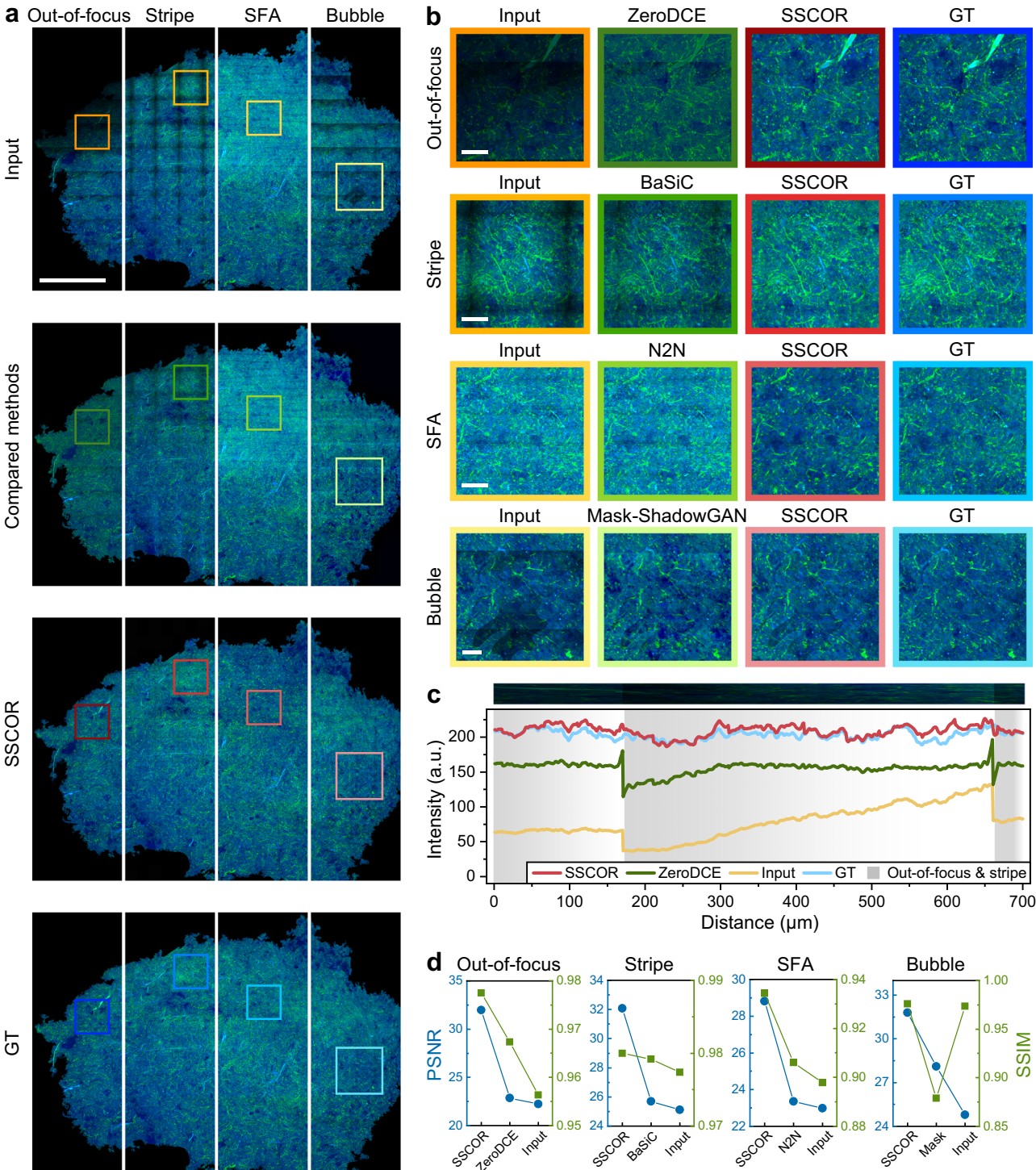

**Fig. 4 | The correction results of special artifacts on stripes. a** The input images are the stripe and three common artifacts synthesized on the original SRS images[18] (ground truth images, GT). Compared with the competing methods, SSCOR not only removes the artifacts and stripes, but also maximally restores the histopathological features on the GT images. Blue and green pseudo-color respectively represent the two channels of stimulated Raman scattering microscope. Scale bar: 1 mm. **b** The enlarged images within the boxes in **a** further proves the restoration ability of SSCOR on detailed features. Scale bars: 100 μm. **c** The representative intensity profiles of out-of-focus magnified areas in **b** demonstrate the capability of SSCOR on artifact removal and stripe correction. **d** Compared with other methods, SSCOR-corrected images have better image quality, as indicated by peak signal-to-noise ratio (PSNR) and structural similarity index (SSIM) of the images in **a**). Source data are provided as a Source Data file.

results of a synthetic stripe image from another H&E dataset, CoNSep[29], were demonstrated and quantitatively evaluated in Supplementary Fig. 9b and 9c. The correction results of other representative synthetic H&E stripe images are presented in Supplementary Fig. 10. As a conclusion, SSCOR is in favor of not only recovering the

ambiguous segmentation of cells, but also improving cell classification. Third, we performed the most common downstream task, automatic cell counting[19], on SRS images of brain tumor with SFA (Fig. 5d). The white dots represent the location and size of the cells. The color-coded markers represent the size of the cells. SSCOR-corrected images

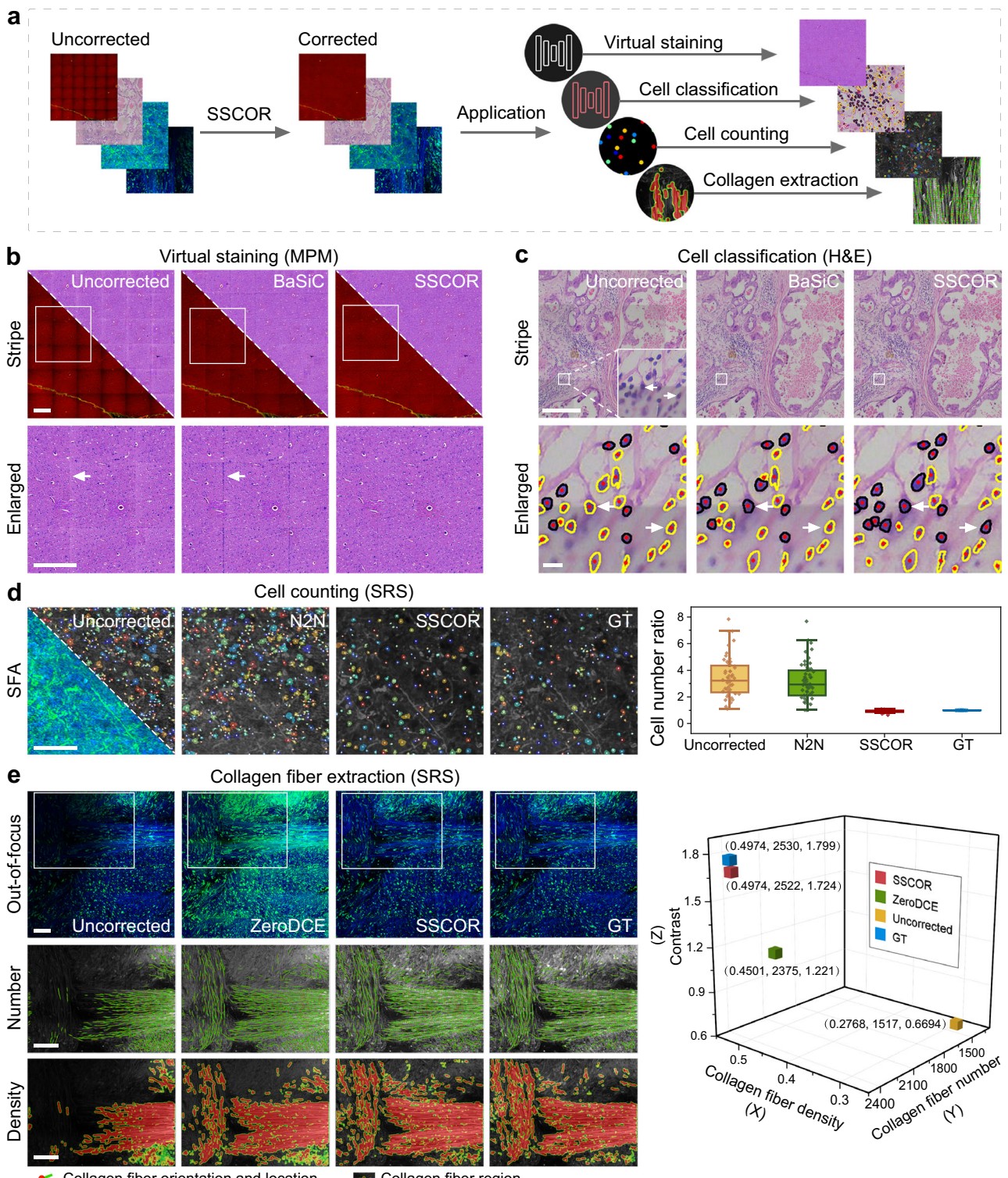

**Collagen fiber orientation and location**     **Collagen fiber region**

significantly reduced the false detection and missed detection of cells compared with N2N method. Furthermore, by quantifying the cell number ratio, the counting results of SSCOR-corrected image was more consistent with GT images. We also performed the cell counting task on bubble artifact and stripe images (Supplementary Fig. 11). In addition, to verify the accuracy of the automatic cell counting algorithm[30], the automatic counting results and manual counting results were compared in Supplementary Fig. 12. Fourth, tumor-associated collagen signatures are associated with the tumor development and disease prognosis[10,28]. In the Fig. 5e, we extracted three

prognosis-related signatures including the collagen fiber number[31], collagen fiber density[31], and contrast feature of collagen fiber[32] in the SRS image with out-of-focus artifact. In the visualization of collagen fiber number and density, ZeroDCE seemed to remove artifacts area, but it also produced incorrect image content. Combined with the quantification of the three extracted collagen signatures, SSCOR showed the best restoration ability for collagen signatures in the out-of-focus artifact image. These results demonstrated that although the stripe or artifact can bring a non-negligible error for the downstream analysis, SSCOR has various stripe and artifact correction capabilities,

**Fig. 5 | SSCOR-corrected images benefit downstream tasks. a** Schematic diagram of SSCOR applications in three downstream tasks. Virtual staining[27], cell counting[30], and collagen fiber extraction[31,32] are performed to verify the significance of SSCOR in typical stripe and artifacts correction. **b** Compared with the corresponding comparison methods, SSCOR-corrected image reduces the misidentification of histopathological features to the largest extent on the tile stitching positions of the virtual stained images. The ground truth (GT) images are the adjacent stained section of the imaging section. The arrows indicate the misgenerated cells caused by stripes. Scale bars: 500 μm. **c** In real H&E images, SSCOR not only recovers more ambiguous cells, but also improves the accuracy of cell classification. The yellow segmented cells represent fibroblasts, and the black segmented ones represent lymphocytes. The arrows indicate the cells misidentified in the uncorrected and BaSiC-corrected images yet correctly identified in SSCOR-corrected images. Scale bar of stripe image: 500 μm. Scale bar of enlarged image: 20 μm. **d** SSCOR-

corrected image significantly reduces the false detection and missed detection of cells. The white dots represent the location of the cells. The color-coded markers represent the size of the cells. The cell number ratio is calculated by the ratio of cell numbers in the uncorrected/corrected image to the GT images ($n = 53$, $n$ represents the quantity of ROIs in scanning fringe artifact images). Box plots indicate median (middle line), 25th, 75th percentile (box) and 1.5 × interquartile range (whiskers). Scale bar: 100 μm. **e** SSCOR-corrected image restores the tumor collagen-associated signatures more realistically, which is reflected in the visualization of collagen fiber number and density. The green lines indicate the orientation of collagen fiber. The red dots indicate the location of collagen fiber. The heatmap is used to visualize the collagen fiber region. The SSCOR-corrected image is also highly consistent with GT image in quantitative comparison of collagen fiber density, number, and contrast. Scale bars: 100 μm. Source data are provided as a Source Data file.

which is able to restore the realistic content of the image hidden in the stripe, and minimize the occurrence of such errors.

Finally, we overviewed the capabilities, advantages, and limitations of SSCOR and all the comparison methods in practical stripe correction scenarios (Table 1). In particular, SSCOR could provide supplementary supports for the off-the-shelf methods, including various types of stripes, less prior information that supplements BaSiC and CIDRE, custom-built system friendly that supplements ZEN, and stripe self-correction that supplements other deep learning methods. In terms of correction time, the prior-free SSCOR takes 45 seconds to correct an image (2881 × 2872 pixels), and the training only relies on the input stitched image itself. Compared with other unsupervised deep learning methods[22–24], SSCOR focuses on fluorescent microscopic image scenarios, the correction time is close to them or even less. Although our method is more time-consuming than the digital image processing-based correction methods such as BaSiC[14], it can be competent for various stripe and artifact correction tasks without raw stitching tiles. As a result, these advantages enable SSCOR-corrected images to provide more precise qualitative diagnosis and quantitative analysis for downstream tasks. Based on the experimentally acquired images and their stripe or artifact types, Table 1 can also be used as a brief guideline to select the appropriate method.

## Discussion

Stripe or shading correction has become an indispensable image post-processing step after microscopic image acquisition. However, the diverse stripes and complex artifacts that often appear in stitched microscopic images have not been effectively addressed. Existing off-the-shelf methods like BaSiC[14] and CIDRE[11] have achieved excellent performance on the uniform stripe correction for microscopic images[11,14,16,19]. They are inspired by the physical process of microscopic image generation, which recover real image intensity by estimating the flat-field illumination variation and the dark-field thermal noise when no light is incident on the sensor. In specific, BaSiC[14] and CIDRE[11] proposed to apply low-rank sparse decomposition and regularized energy minimization to deduce the flat-field and dark-field respectively to correct each image tile. However, many latent factors, e.g., laser source, detector noise, specimen quality, and stability of optical devices, jointly affect the shading patterns of image tiles during practical image acquisition. As a result, the multi-factor influenced shading patterns appear to be inconsistent, or even unpredictable, in a large-scale stitched image, thus appear various types of stripes or artifacts[5–7,10,12,13,33–35]. Thus, the ideal physical process of microscopic image generation simulated by prior methods is not sufficiently capable to model this realistic task. Besides, the raw stitched image information, such as tile size and stitching overlap percentage, are often discarded during data storage and transmission, which poses further challenge for the methods relying on raw tiles.

Previous methods tackle stripe correction from the perspective of shading correction in each raw image tile. In contrast, SSCOR is based

on image patches from post-stitched images. The advantage lies in the following aspects. First, in practice, the stitched information on raw image tiles may not be well retained, so these tile-based methods may not be reliable in such situations, because it is usually impossible to precisely partition tiles. As demonstrated in the Results of high tolerance towards imprecise prior stitched information, compared to the tile-based method, patch-based SSCOR has higher tolerance towards inevitably imprecise tile partition and stripe obliqueness during image cropping and rotating. Next, since each image patch is smaller than a tile, sampling patches easily suffice the training data, which meets the data hunger nature of deep models. Moreover, sampling adequate patches can provide diverse shading patterns rather than constant shading patterns. Thus, patch-based SSCOR also has better generalizability and adaptability for challenging stitched images with various stripes on four datasets with different modalities.

Due to data hunger nature of deep learning models[36–40], large datasets are often the prerequisite for training these models into good form. However, unlike clinical imaging and digital pathology datasets, there are currently few publicly available large datasets on nonlinear optical microscopic images. Therefore, as the major contribution, the proximity sampling strategy is proposed, which enables SSCOR to rely upon one or a limited number of post-stitched images for training, without any patch-wise manual annotation. To do so, SSCOR collects sufficient proximity patch samples on and off the stripes from the same image. Although these samples are unregistered paired, they have similar illumination condition and contextual textures. Besides, the off-stripe patches can serve as the stripe-free references to those on-stripe patches. Thus, it inspires the stripe self-correction process of SSCOR in converting the stripe patches from stitched images into stripe-free patches and eventually merging them together in an unsupervised manner. This self-correction approach provides an effective way to fully exploit the limited training images without requiring any prior stitching information. Hence, driven by our sampling strategy, prior-free SSCOR not only achieves state-of-the-art performance against the off-the-shelf approaches in stripe and artifact self-correction, but also saves a large amount of effort for researchers from the tedious and challenging correction tasks when they demand high-quality stripe-free images for downstream analysis.

In essence, the goal of SSCOR is to correct the shaded patches sampled from stripes. The main challenge lies in the natural blending of the non-uniform shaded regions and the content of patches, as well as multiple latent factors that influence stripe patterns in real-world scenarios, which hinders modeling of physical process. To address this concern, inspired by[25,27,41–44], any image with various stripes can be approximately considered as the feature embeddings from a manifold of high-dimension space that smoothly bridges image domains containing images with and without stripes. By implicitly modeling the manifold space using deep neural network, SSCOR is able to freely translate the images with stripes to the ones without stripes, which circumvents the trouble of modeling latent factors in physical process.

**Table 1 | Comparison of the capabilities of different methods**

| Method | Ability | Applicable Scenarios | Limitations | Correction Time (s) | |
|---|---|---|---|---|---|
| | | | | Image Size (pixel): 6348 × 5376 (13 × 11 tiles) Tile Size (pixel): 512 × 512 | Image Size (pixel): 2881 × 2872 (3 × 3 tiles) Tile Size (pixel): 1024 × 1024 |
| ZEN[15] | Uniform stripe correction | ZEN can correct uniform stripes in the raw image acquired by Zeiss confocal microscope before online stitching. | ZEN demands over 300 tiles per image for satisfactory correction performance, but over-large raw files may cause ZEN software overload and crash. | 3.7 | 3 |
| BaSiC[14] | | BaSiC and CIDRE can handle a stitched image given the information of each tile under uniform shading condition. | BaSiC and CIDRE are designed to process the raw stitching tiles. Once the image has been stitched or cropped, these methods should estimate the tile size first, manually crop tiles, and then correct the uniform shading of each tile. They can hardly handle uniform stripes. | 57.43 | 4.49 |
| CIDRE[11] | | | | 55.95 | 13.5 |
| N2N[23] | Denoising | N2N can denoise the Gaussian noise and Poisson noise in synthetic fluorescence images, as well as the natural noise distribution in real-world noisy images. | Since the noise pattern of scanning fringe artifacts (SFA) is significantly different from natural noises, N2N can hardly recover the original tissue signal from SFA in microscopic images. | 182.63 | 49.5 |
| ZeroDCE[22] | Out-of-focus artifact removal | ZeroDCE can recover out-of-focus-like low-light images, and reduce the influence of the lighting environment on the object color under inadequate lighting conditions. | ZeroDCE is designed to enhance the low-light natural images, while it is not perfectly suitable for restoring the microscopic images. | 288.79 | 59.03 |
| Mask-ShadowGAN[24] | Bubble-like artifact removal | Mask-ShadowGAN can properly remove the bubble-like hard shadows caused by objects blocking and preserve the texture details. | Mask-ShadowGAN uses binary masks to represent shadow regions, so it is better at handling hard (uniform) shadow, but less effective at handling soft (non-uniform) shadow. | 234.3 | 53.89 |
| SSCOR | Various stripe correction and artifact removal | SSCOR can adaptively correct non-uniform, oblique, and grid stripes, as well as remove scanning, bubble, and out-of-focus artifacts, while faithfully preserving the original image content. | At present, SSCOR can only perform near-real time correction. When the image stripe is uniform and the precise tile size is known, BaSiC and ZEN may show better correction effect. | 189.49 | 45.26 |

The correction time of Zen[15], BaSiC[14], and CIDRE[11] were tested by the CPU platform (Intel Core i5). The correction time of SSCOR, Neighbor2Neighbor[23], ZeroDCE[22], Mask-ShadowGAN[24] were tested by the GPU platform (NVIDIA RTX 3090). The training requirements for images were detailed in Supplementary Table 3.

In addition, with the aid of the stripe synthesis network that reciprocally adds the shades to the corrected patches, SSCOR is forced to project the images without stripes back to the counterparts without stripes, so it can effectively regularize the stripe correction process and faithfully preserving the original image content.

In practice, the utilization of SSCOR involves two fundamental steps: (1) rough estimation of stripe positions and (2) selection of normal patches. To optimize the usability and efficiency of our method, we have outlined several guidelines that encompass crucial elements for successful implementation in the following aspects: a schematic diagram of typical stripes and artifacts (Supplementary Fig. 13), the representative sampling cases of stripe and artifact (Supplementary Fig. 1), as well as the corresponding description and sampling strategy (Supplementary Table 4). The provided guidelines empower researchers to effectively sample patches for different stripes and artifacts with minimal manual intervention. Nevertheless, there still remains several limitations in the proposed approach. First, SSCOR offers a semi-automated approach rather than a fully automated one, which necessitates a minor degree of human intervention during the initial phases of training. This intervention is primarily required to approximately determine the positions of stripes and the appropriate patch size. As the future work, we aim to make SSCOR fully automatic while maintaining its generalizability and effectiveness. Second, when handling the uniform stripes, with knowing the prior information on precise tile partition, previous methods like CIDRE[11], BaSiC[14], and ZEN[15] can achieve slightly better performance than SSCOR. The reasons are two folded. On one hand, both uniform stripes with a consistent pattern and precise tile partition provide critical constraints and priors on the estimation of flat and dark fields, which fit the settings of existing tile-based methods. On the other hand, SSCOR does not assume that the patterns of stripes are uniform. Its proximity sampling scheme exhaustively samples a large number of patches that innately contain various stripe patterns. Thus, it sacrifices the model's robustness over uniform stripe, but it increases the generalizability for various non-uniform stripes. Third, SSCOR may have difficulty distinguishing between the heavy stripes and the original low-intensity content in the image, potentially resulting in incorrect corrections. In these cases, the model may either hallucinate erroneous details to compensate the degraded image content caused by heavy stripes, or choose to neglect the stripes due to treating them as part of the original image content. Fourth, both the adversarial self-training stage and local-to-global correction stage of SSCOR require GPU-based computation resources and cost more running time than previous methods. So, SSCOR can hardly be applied to image sequences or video in the current form. The adversarial self-training is performed offline, which usually demands sufficient time to train a robust model, but the offline training time may not significantly affect the downstream applications. As the future work, we will focus on the acceleration of local-to-global correction stage. To do so, the structure of SSCOR can be further optimized so as to deploy on the edge devices[45–48] and the local-to-global correction can be further accelerated using parallel computation[49,50]. Furthermore, since SSCOR was implemented based on PyTorch, it can be easily transplanted to different platforms such as portable computing devices or cloud-based AI inference engines, and we are going to implement it as plug-in of open-source platforms like Deep-ImageJ[51]. More importantly, the code and pretrained models of SSCOR are ready to release in public, so that they can be continuously updated and extended to more data, which can benefit more imaging and biomedical fields.

To sum up, we propose a deep learning-based stripe self-correction method, SSCOR. In principle, our proposed proximity sampling scheme and adversarial reciprocal training paradigm enable SSCOR to utilize normal patches as reference to correct their anomaly counterparts. As a consequence, SSCOR can only rely on input stitched image itself to adaptively correct non-uniform, oblique, and grid stripes, as well as remove scanning, bubble, and out-of-focus artifacts, while faithfully preserving the original image content. Through the comprehensive experiments on three fluorescence datasets of different modalities, SSCOR achieves the state-of-the-art performance for stripe correction and artifact removal. Comparing to off-the-shelf retrospective approaches, the prior-free SSCOR does not require any physical parameter estimation, patch-wise manual annotation, and raw stitched information in the correction process, which is a researcher-friendly approach that provides supplementary support for the digital image processing-based correction methods. In addition, it is also demonstrated that SSCOR significantly improves the precision of downstream analysis for common applications. As the future work, the efficiency of SSCOR can be further improved for use in edge devices or as plug-in of open-source platforms, which can benefit more imaging and biomedical communities by embracing more modality data.

## Methods

### Ethical statement

All anonymous tissue collections for retrospective study of MPM imaging were conducted under a protocol approved by the Institutional Review Boards (IRB) of Fujian Medical University Union Hospital (2020-085). For the user study, all participants provided written informed consent prior to participating in the image quality questionnaire. This user study was conducted under a protocol approved by the Institutional Review Boards (IRB) of Fujian Normal University (IACUC-20230039).

### Microscopy datasets

The following microscopic datasets were used in this study: (1) the MPM dataset that contains breast images with non-uniform stripe, liver cancer images with grid stripe, and cerebral vascular malformation images with uniform stripe; (2) the mouse brain fluorescence dataset[17] that includes the images with oblique stripe; (3) the images from SRS dataset[18] were used to synthesize various artifacts and stripes; (4) the images from H&E dataset[29,52] were used to synthetic stripes.

Fluorescence, SRS, and H&E are publicly available datasets. For the acquisition of MPM dataset, we used a multiphoton imaging system consisting of a commercial confocal microscope (Zeiss LSM 880 META, Jena, Germany) and an external mode-locked Ti: sapphire laser (140 fs, 80 MHz)[10,26], in which the excitation wavelength was 810 nm and the average power of the samples was 30 mW. Tissues used for imaging were extracted during surgery and prepared as formalin-fixed and paraffin-embedded samples. The samples were sliced consecutively into 5-μm sections using a microtome. The imaged sections were deparaffinized by alcohol and xylene prior to imaging. The adjacent sections were stained with H&E for histopathological comparison. A comparison of these datasets was detailed in Supplementary Table 5.

### Network architecture

The unpaired images $X$, $Y$, which are the anomaly patch domain and normal patch domain, respectively. In order to restore the anomaly patches and preserve the original content, SSCOR mainly composes of four sub-networks: (1) the stripe correction network $G_C$ that aims to correct the input anomaly patch; (2 and 3) two discriminator networks, $D_X$ and $D_Y$, that utilizes the unpaired patch as reference to evaluate the quality for the corrected patch and synthetic patch; (4) the stripe synthesis network $G_S$ that adds shades to the corrected patch and thus compares it with the original anomaly patch, which enables to preserve the image content from over-correction. Supplementary Fig. 14 depicts the detailed structures and the relationship of the networks in the proposed framework.

In concrete, the stripe correction network of SSCOR is based on an encoder-decoder network structure, which is composed of downsampling layers, intermediate layers, and upsampling layers. First, the

downsampling layers are essentially two strided convolutional layers for extracting low-level visual features of the input patch. Next, in the intermediate layers, there follows nine stacked residual blocks[36] to extract high-level semantic features for describing visual histological components. Last, there are two upsampling layers implemented by strided convolution as well, which are used to integrate features extracted from previous layers and reconstruct them to the corrected patch with the original dimension. Besides, the discriminator network is composed of five convolutional layers, in which the first four layers are the convolutional layers for extracting deep visual features and the last convolutional layer serves as a classifier. Moreover, the structure of the stripe synthesis network is identical to that of the stripe correction network.

SSCOR employs the stripe correction network $G_C$ and stripe synthesis network $G_S$ to learn the mapping between the image domain of any anomaly patch $X$ and the image domain of any normal patch $Y$. During training, the overall loss function $L$ can be defined below:

$$L = L_{adv} + \lambda L_{cons}, \qquad (1)$$

where $L_{adv}$ refers to the adversarial loss that consists of the losses for training the stripe correction network $G_C$ and stripe synthesis network $G_S$, as well as the losses for training discriminator networks $D_X$ and $D_Y$. Formally, the adversarial loss is composed of four terms,

$$L_{adv} = L_{adv}^{G_C} + L_{adv}^{G_S} + L^{D_Y} + L^{D_X}. \qquad (2)$$

Besides, $L_{cons}$ refers to the content consistency loss that aims to enhance the capability of preserving the original image content for the stripe correction network $G_C$ and stripe synthesis network $G_S$. $\lambda$ is a constant set as 10 to balance these two loss terms.

Adversarial losses. During training, an anomaly patch and its associate normal patch in proximity will be passed to the discriminator network to distinguish if they are normal. If the anomaly patch was not well corrected, the network could easily discriminate them. Hence, the objective is to enhance the quality of the anomaly patch to its maximum extent, thereby maximizing the difficulty of discrimination. For $G_C$ and $G_S$, their losses can be formulated as:

$$L_{adv}^{G_C} = \mathbb{E}_X \left[ (1 - D_Y(G_C(X)))^2 \right], \qquad (3)$$

$$L_{adv}^{G_S} = \mathbb{E}_Y \left[ (1 - D_X(G_S(Y)))^2 \right], \qquad (4)$$

where $\mathbb{E}_X(\cdot)$ and $\mathbb{E}_Y(\cdot)$ compute the expected values of the distribution of image domain $X$ and $Y$, respectively. These two losses leverage discriminators to evaluate the quality of stripe correction and stripe synthesis. In other words, the harder it is for the discriminators to distinguish between the processed patches and the reference raw patches, the better the generated results will be. On the other hand, the adversarial losses for training the discriminator networks $D_Y$ and $D_X$ that are dedicated to strengthen the ability of discriminating the processed patches and the reference raw patches, can be written as below:

$$L^{D_Y} = \mathbb{E}_Y \left[ (1 - D_Y(Y))^2 \right] + \mathbb{E}_X \left[ D_Y(G_C(X))^2 \right], \qquad (5)$$

$$L^{D_X} = \mathbb{E}_X \left[ (1 - D_X(X))^2 \right] + \mathbb{E}_Y \left[ D_X(G_S(Y))^2 \right]. \qquad (6)$$

Content consistency loss. In order to preserve the original image content without being deteriorated during generation, the consistency constraint is achieved by reciprocally reconstructing, which requires $G_C$ and $G_S$ to be jointly learned. The content consistency loss essentially introduces two reciprocal processes: "destriping-shading" and "shading-destriping". Specifically, in the "destriping-shading" process, we first remove stripes from the input anomaly patch, and subsequently synthesize the stripes back to their corrected form, obtaining a synthesized patch. By constraining the synthesized patch to be similar to the input anomaly patch, we are able to maintain image content quality during network inference and achieve content consistency. Furthermore, we also implemented a reciprocal "shading-destriping" process on an input corrected patch to further reinforce content consistency. Thus, the content consistency loss can be formulated as:

$$L_{cons} = \mathbb{E}_X[\| G_S(G_C(X)) - X \|_1] + \mathbb{E}_Y[\| G_C(G_S(Y)) - Y \|_1]. \qquad (7)$$

In particular, the first term aims to guarantee that an anomaly patch $X$ after being successively processed by $G_C$ and $G_S$ should be consistent to itself in a reciprocal manner. Similar to the first term, the second term let a normal patch $Y$ be consistent to itself after being processed by $G_S$ and $G_C$. This reciprocal process confines $G_C$ and $G_S$ to generate desired results and preserves the original content.

### Network implementation details

The network architecture of SSCOR is illustrated in Supplementary Fig. 14, where each cube represents a multi-channel feature map. The respective label (e.g., $256 \times 256 \times 64$) under each cube indicates the spatial dimensions and channel information of the corresponding feature map. The detailed description of our framework is elucidated as below.

In the stripe correction network, the first three layers are implemented as downsampling layers, using convolution (Conv), Instance Normalization (IN), and Rectified Linear Unit (ReLU). For input patches with dimensions of $256 \times 256 \times 3$, a convolutional layer is first applied to increase the feature depth to $256 \times 256 \times 64$. The feature map is then downsampled by two in the next two downsampling layers, while the number of channels is doubled by the convolution with stride 2. This process results in a $128 \times 128 \times 128$ feature map capable of extracting low-level visual features of the input patch. To extract high-level semantic features for describing visual histological components, nine stacked residual blocks are employed. These blocks incorporate shortcut connections to facilitate the information flow during network training. The first two upsampling layers are realized by a transpose convolution with stride 2, followed by IN and ReLU. The final upsampling layer is realized by a convolutional layer with stride 1, followed by Hyperbolic Tangent (Tanh) activation. In addition, the discriminator of our model employs a relatively shallow CNN architecture, with the last convolution layer producing a single-channel feature map for a patch to be classified as either normal or anomaly.

For training SSCOR, we utilized the Adam solver[53] with a learning rate of 0.0002 and exponential decay rates for the first and second moment estimates set to 0.5 and 0.999, respectively. During training, the number of epochs is set to 200, with a fixed learning rate in the first 100 epochs and a linear decay to zero over the next 100 epochs. Note that, we did not utilize any pre-trained weights during the training process. The training procedure is stable even with different random initializations (Supplementary Fig. 15). The hyper-parameters of the sampling strategy are described in the Supplementary Table 6. The optimal step size for patch sampling is determined according to the experimental results in Supplementary Fig. 16. During inference, the procedure of the local-to-global correction is illustrated in Supplementary Fig. 17. We set the step size of sliding-window as half of the patch size or a fixed value of 100 pixels.

## Proximity sampling strategy

Proximity sampling strategy serves as one of the most critical components in SSCOR. For any stitched images with different stripes and artifacts, the sampling strategy can be summarized as: (1) define normal and abnormal regions; (2) iteratively sample an anomaly patch from abnormal region and a normal patch from normal region in proximity to pair with the normal patch. Specifically, the proximity sampling strategy consists of the following steps. First of all, the partition of tiles can be manually annotated by users even without expertise and the rough locations of stripes can be initially determined. Although the manual partition inevitably be imprecise due to the unknown width or slight oblique of stripes, SSCOR can tolerate these mild errors. Next, based on the rough locations of the stripes, image patches can be randomly sampled accordingly. Specifically, the patches sampled around the stripes are considered as anomaly patches, while those far away from the stripes are normal patches. In practice, the dimension of the sampled patches is predefined as the image regions smaller than that of the original image tiles, which enables SSCOR to sample a large number of normal/anomaly patches while obtaining sufficiently diverse stripe patterns. For training SSCOR, each anomaly patch needs to pair with a normal patch. To ensure the quality of image restoration, the paired normal and anomaly patches need to be located in a proximity. To do so, given a sampled anomaly patch, the nearest adjacent normal patch is chosen. Besides, for an image containing not only stripes but also specific artifacts, the sampling strategy is slightly different. Concretely, the anomaly patches are mainly sampled from the region with artifacts, while the corresponding normal ones are obtained from the closest normal image region.

To improve the usability of SSCOR for correcting various types of stripes or artifacts, we drew the schematic diagrams for illustrating six typical stripes and artifacts (Supplementary Fig. 13). Additionally, a comprehensive description of the various types of stripes and artifacts, along with their corresponding sampling strategies, is provided in Supplementary Table 4. Detailed settings of the proximity sampling strategy applied to the experimented images are outlined in Supplementary Table 6. Moreover, the ablation studies on the effectiveness of sampling strategy, adversarial reciprocal-training, local-to-global correction were conducted, as described in Supplementary Note 4 and Supplementary Fig. 18.

## Benchmarks

For comparison, we introduced several off-the-shelf methods including CIDRE[11], BaSiC[14], and ZEN[15]. Among these methods, as retrospective methods, CIDRE[11] and BaSiC[14] proposed optimization based approaches to accomplish tile-based shading correction. ZEN[15] is a commercial software from Zeiss, which requires a reference image generated from image tiles to perform shading correction. All of them can be applied to the task of uniform stripe correction. In addition, some latest deep learning-based methods, like ZeroDCE[22], Neighbor2Neighbor[23], and Mask-shadowGAN[24], were involved in experiments. As latest representative models for image enhancement, they were originally designed for various tasks including low-light enhancement, image deshadow, and image denoising. They can be adapted to the tasks related to artifact removal. For more detailed information on these comparison methods, please refer to Supplementary Note 5.

## Evaluation protocols

We adopted multiple metrics to evaluate the correction quality through comprehensive experiments. For microscopic images without the available ground truth (GT), two metrics were used to evaluate the correction quality. The first metric is the intensity profile, which measures the discrepancy between the corrected and uncorrected image or ground truth. The intensities were averaged within the ROIs in the direction parallel to the stripes. The intensity profiles in stripe and non-stripe areas can respectively reflect the abilities for removing

stripe and preserving the original image features. The second metric, inverse coefficient variation (ICV), is used to quantitatively evaluate the correction effect, which can be calculated as follows:

$$ICV = R_a/R_{sd} \tag{8}$$

where $R_a$ refers to the signal intensity of microscopic image, which is calculated by the averaged pixel intensities of an ROI. $R_{sd}$ is used to estimate the degree of intensity fluctuation in the stripe area, which is calculated by the standard deviation of pixel intensities. A larger ICV corresponds to the flatter intensity profile and quantitatively indicates a better correction quality. Notably, ICV is sensitive to the sudden changes in the signal intensity of image content. The black background should be avoided when selecting the ROI, otherwise the ICV will be disturbed by the image content signal.

For synthetic data with GT, we additionally adopted the full-reference quality metrics, PSNR and SSIM, to assess the image quality and structural similarity between corrected image and GT.

In order to verify the contribution of SSCOR-corrected images to downstream applications, we applied virtual pathological staining, automatic cell counting, and collagen signatures extraction to quantify the ability of SSCOR for preserving the original tissue characteristics, respectively. The virtual staining was achieved by an unsupervised content-preserving transformation network (UTOM)[27]. The cell segmentation and classification were performed by Hover-Net[29]. The cell numbers were counted using the software package CellProfiler 4.2.1[30]. The number and area of collagen fibers were extracted and qualified by CurveAlign v4.0[31]. The contrast feature of collagen fibers calculated by a gray-level co-occurrence matrix-based algorithm[32] (Matlab vR2021b, the MathWorks Inc.).

## Hardware and software

All experiments were conducted on a workstation with a NVIDIA RTX 3090 GPU. Each model was trained on a single GPU using the PyTorch library (v1.7.1). Cropping patches, synthetic image generation and quality evaluation were performed in Python (v3.7).

## Statistics & reproducibility

All statistical analyses were performed using GraphPad Prism (version 9.0.0, GraphPad Software, San Diego, California USA). To compare three or more groups, a one-way analysis of variance (ANOVA) was utilized, followed by Tukey's multiple comparisons test. If the normality test failed, Kruskal-Wallis test and Dunn's multiple comparisons test were used. All box plots indicate median (middle line), 25th, 75th percentile (box) and 1.5 × interquartile range (whiskers). The significance level is displayed as asterisks, and $P < 0.05$ was considered statistically significant (*$P < 0.05$, **$P < 0.01$, ***$P < 0.001$, ****$P < 0.0001$; ns, not significant). No statistical method was used to predetermine sample size.

Under the unsupervised learning settings, for each input image, all the models were trained once per set of hyper-parameters, and tested on the same image. With different initial weights, the proposed SCCOR eventually achieved similar results. To facilitate reproduction, we release the trained model, codes, and source data at https://github.com/lxxcontinue/SSCOR.

## Reporting summary

Further information on research design is available in the Nature Portfolio Reporting Summary linked to this article.

## Data availability

The MPM data of non-uniform/grid stripes used in this study are available at our github repository https://github.com/lxxcontinue/SSCOR. The fluorescence dataset[17] used for oblique stripes are available at http://brainarchitecture.org/. The SRS dataset[18] used for special

artifacts synthesis are available at https://dataverse.harvard.edu/dataset.xhtml?persistentId=doi:10.7910/DVN/EZW4EK. The CoNSep dataset[29] used for stripes synthesis are available at https://warwick.ac.uk/fac/cross_fac/tia/data/hovernet/. The MoNuSAC dataset[52] used for stripes synthesis are available at https://monusac-2020.grand-challenge.org/Data/. Source data are provided with this paper.

## Code availability
The codes, pretrained models, and relevant resources of the proposed SCCOR are publicly released with a detailed guide at https://github.com/lxxcontinue/SSCOR.

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

## Acknowledgements

We would like to thank Lianhuang Li and Xiahui Han for their support in the data acquisition of multiphoton images. This work was support by the National Natural Science Foundation of China (62005049 to S.W., 62072110 to W.L., and 82171991 to J.C.), the Natural Science Foundation of Fujian Province (2020J01451 to S.W. and 2022J01216 to J.C.), the Special Funds of the Central Government Guiding Local Science and Technology Development (2020L3008 to J.C.), and Fujian Major Scientific and Technological Special Project for "Social Development" (2020YZ016002 to J.C.).

## Author contributions

S.W., W.L., F.H., and J.C. designed and directed the study; X.L., Q.L., and Y.S. set up the SSCOR framework; Y.L., X.S., and Y.X. performed quantitative analysis; S.W. and X.H. performed multiphoton imaging experiments; D.K., X.W., and R.L. provided specimen; S.W., W.L., X.L., and Y.L. wrote the manuscript. H.T. and J.C. reviewed and revised manuscript.

## Competing interests

The authors declare no competing interests.
