## [Peer Review File · Nature Communications]

A deep learning-based stripe self-correction method for stitched microscopic imagesREVIEWER COMMENTS

Reviewer #1 (Remarks to the Author):

This paper uses the depth learning method to improve the strip self-correction of the spliced microscopic image. The paper involves many different strip conditions, artificial violations, and uneven illumination. By comparing with some relevant advanced research methods, the advantages of the provided methods are demonstrated. At the same time, the data acquisition and experimental setup are introduced in detail, which is a valuable research. However, there are some problems in the paper that need to be improved. At the same time, some questions need to be further explained and demonstrated.

The impact of microscope instrument and amplification resolution on grid stripes needs to be considered. Can we ensure that grid stripes are less than 128 pixels?

If artifacts and other interferences are added into stripes, it will be very difficult to select normal image blocks. Therefore, an interactive strategy with experts involved is needed. At present, it is very difficult to achieve the full-automatic selection of normal image blocks under any imaging conditions and any instrument. It is not feasible and cannot achieve full-automatic selection.

Supplementary Figure 8 is not be cited in the Supplementary Note. Some symbols in Figure 8 need to be explained. At the same time, what is the meaning of nine times 9x ?

The network architecture here is somewhat simple, and attention, mechanism and multi-scale mechanism are not involved. The image enhancement effect needed to be explained further.

In the stage of strip correction and synthesis, if the artificial interferes of correction discipline, then there will be problems in the content consistency judgment of image blocks. How to ensure the accurate maintenance of image features?

In equation 7, there should be symbol errors and missing.

It is very important to evaluate the quality of image restoration. The application of lcv here should not be a very good choice.

In short, it is very difficult to realize automatic correction of microscopic image stripes so far, because it involves different imaging conditions and equipment as well as parameters

selection and amplification resolution, etc. Second, there are some problems in the network structure design, and the handling of complex situations should be further explained. The selection of normal image blocks and image quality evaluation are very important and should be further studied.

Reviewer #2 (Remarks to the Author):

The manuscript entitled, “A deep learning-based stripe self-correction method for stitched microscopic images”, the authors develop a method to remove the stripping artifact and other anomalies from large, tiled images. The use of paired normal and anomaly patches to reciprocally train the ablation studies on the effectiveness of sampling strategy, adversarial reciprocal training, and local-to-global correction are a strength of the manuscript.

Considering the increasing dependence on imaging and image analysis in biological research this is timely and important. Tools to improve the accuracy and reproducibility of imaging data analysis will be beneficial across multiple disciplines. Overall, the manuscript is clear and logical with strong validation of the methods. There are only a few concerns that should be addressed before publication.

1. The images chosen for the figures are relatively even in GT fluorescence signal across the tissue. It is unclear if at least one channel of fluorescence needs to be fairly homogenous across the tissue/sample for effective correction. To demonstrate that SSCOR does not over or under correct it would be good to test on an image that has GT heterogeneity in fluorescent signal. This is quite important for analysis where the intensity of a particular marker is a comparison of low vs high expression.

Further, would the location of a strip and subsequent correct impact the accuracy of heterogenous fluorescence intensity? For example, does strip correction in a region of low-intensity lead to overestimated signal or conversely does strip correction in a region of high fluorescence intensity underestimated the signal?

2. Minor comment – When discussing collagen quantification, the analysis software (CT-Fire) quantifies the number and density of collagen fibers, not total collagen. The word ‘fiber’

should be included in the text.

3. Figure 4D. It is unclear what the 4 different graphs represent. Does each represent a different field of view or region of interest within the tiled image?

Reviewer #3 (Remarks to the Author):

Summary:

This paper presents an unsupervised patch-wise deep learning method for correcting stripes and other artifacts for stitched microscopic images. The deep learning-based correction is performed in a patch-wise manner, and the whole image can be processed in a sliding window manner. The authors propose the proximity sampling strategy to sample normal and abnormal patches in the given image and adopt the unsupervised CycleGAN strategy to train the model. The authors conduct experiments on three datasets and demonstrate the effectiveness of the proposed approach on stripe corrections and artifact removal.

Strengths:

1. The paper provides a solution for unsupervised stripe correction and artifact removal, which is an important problem in microscopic image processing, considering the difficulty of acquiring paired/annotated datasets.
2. The paper is well written, and the authors present their approaches and results clearly.
3. The authors validate the benefits of SSCOR for various downstream tasks.

Weakness:

1. The impact/novelty is limited. Deep learning-based stripe/artifacts correction is not a new topic. As mentioned in the paper, there are some supervised learning-based approaches for

that. One of the key points is the unsupervised learning manner. While unsupervised learning is achieved with the well-studied CycleGAN strategy. From the technical aspect, the novelty is incremental.

2. The motivation/description/rationale of some key components in the methodology part is unclear.

a) The proposed approach adopts patch-wise correction and uses the "local-to-global" strategy to correct the whole image. However, the authors should clarify how they ensure the consistency/smoothness of different patches when merging the patch-based results. Otherwise, this manner may introduce secondary stripes in the image.

b) The proposed approach heavily relies on the sampled normal/abnormal patches, while this sampling part involves too much human intervention from the description in the supplementary material. For example, it needs humans to manually annotate the partition of tiles, determine the dimension of patches, and also the manner to get the normal patches(with human-defined rules). As we need to conduct this sampling/training procedure for each dataset (or image), how do you ensure the robustness and generalization of the proposed framework for new testing cases? Especially for the sampling of normal patches in the images with artifacts, the adopted strategy is quite objective. Can you quantitatively describe the adopted strategy?

c) One of the key points for training deep models is avoiding underfitting or overfitting. If the model is underfitting, it cannot correct the strips well. If the model is overfitting, it maybe generates some over-smooth results and discards some important content details. This point is more important in the proposed approach, as we need to train one model for each testing data. Unfortunately, the authors did not describe/discuss this point in detail.

d) In the discussion part, the authors mentioned that previous methods like CIDRE, BaSiC and ZEN may achieve slightly better performance than SSCOR when knowing the prior information on title partition. In the proposed framework, we also need to roughly annotate the portion of tiles. In this case, what is the advantage of the proposed approach?

3. For the presented results (Figs.2&3&), there are still many stripes in the corrected image, although the proposed approach achieves better results than other baseline methods. This may indicate that the proposed approach may have limitations in correcting heavy stripes.

Other comments:

1. Please clarify the robustness of the proposed proximity sampling strategy. Can it "correctly" sample normal regions under different situations? For example, there are artifacts near the stripes.

2. Please discuss more details about the training procedure, including whether it is difficult to train the GAN-based framework, the stability of the training procedure, and whether any pretrained weights are used in the model.

3. In Fig. 5c, please show the correction results of BaSiC and SSCOR. For the enlarged patches, it seems that the correction effect for these two patches is mild. I am not sure why this correction can improve the segmentation/classification performance.

Reply Letter

We appreciate the time and efforts from the reviewers. We have carefully considered all the comments and suggestions raised by the reviewers and have addressed them in our point-by-point response. The corresponding changes to the resubmitted manuscript have been highlighted in red.

Reviewer #1:

This paper uses the depth learning method to improve the strip self-correction of the spliced microscopic image. The paper involves many different strip conditions, artificial violations, and uneven illumination. By comparing with some relevant advanced research methods, the advantages of the provided methods are demonstrated. At the same time, the data acquisition and experimental setup are introduced in detail, which is a valuable research. However, there are some problems in the paper that need to be improved. At the same time, some questions need to be further explained and demonstrated.

Response: Thank you very much for your overall positive comments on our study. The comments are valuable for improving the quality of our paper. In our revised manuscript, we have responded to all your points below. The corresponding changes to the resubmitted manuscript have been highlighted in red.

1. The impact of microscope instrument and amplification resolution on grid stripes needs to be considered. Can we ensure that grid stripes are less than 128 pixels?

Response: We appreciate the reviewer's comment and would like to further clarify the term "grid stripe". The grid stripe refers to the presence of both horizontal and vertical stripes that are visible in the stitched image, creating a fine grid-like pattern. The pixel width of the grid stripe is related to the amplification resolution of the stitched image. We would like to note that the stripe width of 128 pixels only pertains to the grid stripe image presented in Fig. 2a of our manuscript.

To illustrate the patterns of grid stripes, Supplementary Fig. 15 provided a schematic diagram showcasing the grid stripe, along with other typical stripes and artifacts. Moreover, to improve the usability of SSCOR for correcting various types of stripes or artifacts, their corresponding descriptions and sampling strategy guidelines provided in Supplementary Table 5. We hope

these additions will provide more detailed guidance to instruct users for using our model for different types of stripe correction and artifact removal.

In the revision, we have revised the Method Section 5.3 (Proximity Sampling Strategy) as below, which also adds a reference to Supplementary Fig. 15 and Supplementary Table 5.

"Proximity sampling strategy serves as one of the most critical components in SSCOR. For any stitched images with different stripes and artifacts, the sampling strategy can be summarized as: 1) define normal and abnormal regions; 2) iteratively sample an anomaly patch from abnormal region and a normal patch from normal region in proximity to pair with the normal patch. Specifically, ...

... To improve the usability of SSCOR for correcting various types of stripes or artifacts, we drew the schematic diagrams for illustrating six typical stripes and artifacts (Supplementary Fig. 15). Additionally, a comprehensive description of the various types of stripes and artifacts, along with their corresponding sampling strategies, is provided in Supplementary Table 5. Detailed settings of the proximity sampling strategy applied to the experimented images are outlined in Supplementary Table 6. ..."

Supplementary Figure 15. Schematic diagram of typical stripes and artifacts. We show the diagrams of non-uniform stripe, grid stripe, oblique stripe, bubble-like artifact, scanning fringe artifact, and out-of-focus artifact. The sampling strategy of the stripes and artifacts are outlined in Supplementary Table 5.

Supplementary Table 5. Description of stripe and artifact types and the corresponding sampling strategy.

Type	Description	Causes	Sampling strategy
Non-uniform stripe	The stitched image presents non-uniform stripe pattern, in which each tile exhibits diverse shading patterns.	Multiple factors like the condition of specimen or microscope result in non-uniformity.	Since the exact positions of stripes in stitched images are unknown, we assume that the stripes are uniformly distributed and perpendicular to the image boundaries. This assumption applies even to images with oblique stripes. The sampling steps are as follows:
Grid stripe	The stitched image presents horizontal and vertical stripes that resemble a grid pattern, in which each tile shares the same shading pattern.	The shading of each tile is caused by uneven illumination.	 1. Estimate the position of each stripe in the stitched image based on image width, height, and the number of stripes. 2. Sample anomaly patches along each estimated horizontal or vertical stripe using a sliding window approach.
Oblique stripe	The vertical or horizontal stripes of stitched images present slightly oblique.	Pathological tissues are commonly adhered askew on glass slide for microscopic examination, which may necessitate proper rotation of the acquired image to meet downstream application requirement. This rotation can lead to the presence of oblique stripes in the image.	 3. The striped area covered by the sampling is considered the abnormal region, with the remaining non-striped area being the normal region. 4. Once an anomaly patch is sampled, the corresponding normal patch can be obtained from the nearby non-striped areas using the proposed strategy (please refer to Sec. 5.3 in Method).
Bubble-like artifact	The artifact appears bubble-like irregular shape with uneven shading.	The artifact is essentially local tissue moisture loss caused by laser photodamage or the unsealed specimen.	 1. Localize the abnormal region through user interaction, where the user can draw a coarse region around the artifacts to define the abnormal region. The remaining image region can be considered as normal region.
Scanning fringe artifact	The artifact typically appears throughout the entire image, accompanied by a large area of noise.	The artifact is often produced by a high-speed galvo-resonant scanning imaging system.	 2. Sample anomaly patches within the abnormal region, while normal patches can be sampled from adjacent normal regions outside the abnormal region.
Out-of-focus artifact	The artifact usually locates in the corner of the stitched image, which appears significantly reduction of the signal intensity within shading.	Due to the unevenness of the boundary area of the tissue sample, the microscope is out of focus on these areas.	
Co-existed artifacts and stripes	The aforementioned artifacts overlap with the stripes in the stitched image.	Refer to the above reasons with regards to the type of stripe and artifact.	 1. Apply the aforementioned strategies jointly to define abnormal regions encompassing both stripes and artifacts. 2. Sample anomaly patches from abnormal regions, while normal patches are sampled from nearby normal regions.

2. If artifacts and other interferences are added into stripes, it will be very difficult to select normal image blocks. Therefore, an interactive strategy with experts involved is needed. At present, it is very difficult to achieve the full-automatic selection of normal image blocks under any imaging conditions and any instrument. It is not feasible and cannot achieve full-automatic selection.

Response: We thank the reviewer for raising this point. We would like to clarify that in our manuscript, the proposed unsupervised method for adaptively handling various stripes does not imply that it enables full-automatic selection. Rather, "unsupervised" indicates that our model is trained without using patch-wise manual annotation data during the correction process. Furthermore, "adaptive" refers to the versatility of our model towards various types of stripes, without the need for specially designing the model.

Compared to the off-the-shelf methods, our method focuses more on resolving various non-uniform stripes and special artifacts. Therefore, we acknowledge that achieving a fully automated strategy without any manual input from experts can be challenging at the current stage. For instance, in practice, it can be extremely difficult to heuristically design a fully automated, yet effective model that can precisely identify the artifact area that the user desires to restore.

Despite this difficulty, our proposed method achieves good performance with little manual intervention, and without the need for high-precision sampling. This was shown in the Results 2.3 (Fig. 3), which is attributed to the high tolerance and robustness of our method to imprecise prior information, such as tile size and tile overlap ratio. On the other hand, based on our experience, the rough annotation of normal and abnormal regions from different stripes and artifacts is not a tedious task for users.

Finally, to increase the usability and efficiency of our method, we added interactive guidance to provide users with correction assistance, including a schematic diagram of typical stripes and artifacts (Supplementary Fig. 15), along with the corresponding description and sampling strategy (Supplementary Table 5). Combined with Supplementary Fig. 1 (Representative sampling cases of stripe and artifact), these three supplements have improved the implementation details on how to sample patches for different stripes and artifacts with minimal manual effort. Please refer to the Supplementary Fig. 15 and Supplementary Table 5 from the responses to the first question. Besides, we have outlined the sampling strategy settings used for the representative images in our paper in Supplementary Table 6, which is shown below.

Supplementary Table 6. Proximity sampling on representative cases.

Stripes/Artifacts types	Specimen	Resolution(pixel)	Reference	Tiles	Sample strategy			
					Step size (pixel)	Patch size (pixel)	Sample number	
Non-uniform	Cerebral vascular malformation	2881×2872	Supplementary Table 3	3×3	64	256×256	166	
		6387×6387	Supplementary Fig. 7b	7×7	128		576	
	Breast cancer	3430×3430	Supplementary Fig. 3a	7×7	128	256×256	198	
		6348×5376	Fig. 2a	11×13	128		633	
		11109×9122	Fig. 5c	5×5	256		512×512	316
Stripes	Oblique	4172×3737	Fig. 3b	9×10	64	256×256	210	
		4354×3086	Fig. 2a	8×11	64		220	
	Grid	5000×3750	Supplementary Fig. 3c	11×13	128	128×128	231	
		4403×4403	Fig. 2a	9×9	128		544	
Out-of-focus		7350×5390	Fig. 4	11×15	128		735	
Artifacts	Bubble	Glioblastoma	7350×5390	Fig. 4	11×15	128	256×256	735
	SFA		7350×5390	Fig. 4	11×15	128		470

3. Supplementary Figure 8 is not be cited in the Supplementary Note. Some symbols in Figure 8 need to be explained. At the same time, what is the meaning of nine times 9x ?

Response: Thanks for the careful review. In response to the reviewer's comments, the notation "9×" indicates the cascading of nine identical residual blocks. In the revision, we have provided a more complete description of the symbols used in the caption of Supplementary Fig. 14 (i.e., Supplementary Fig. 8 in the original submitted manuscript). Additionally, we have added the description of our network architecture in Supplementary Note 4: Network implementation details. Supplementary Fig. 14 was cited in the Supplementary Note 4.

Supplementary Figure 14. Visualization of network architecture. **a** The stripe correction network consists of downsampling layers, intermediate layers, and upsampling layers. The discriminator network consists of five convolutional layers. **b** The diagram shows the image data flows during the adversarial self-training stage of SSCOR. Each cube in **a** represents a multi-channel feature map, and the label below each cube indicates the spatial dimensions and channel information specific to that feature map. Conv, convolution layers; IN, Instance Normalization; ReLU, Rectified Linear Unit; Tanh, Hyperbolic Tangent; Shortcut connections, the information flow skipping one or more layers; 9×, the cascading nine identical residual blocks.

"Supplementary Note 4: Network implementation details

The network architecture of SSCOR is illustrated in Supplementary Fig. 14, where each cube represents a multi-channel feature map. The respective label (e.g., $256 \times 256 \times 64$) under each cube indicates the spatial dimensions and channel information of the corresponding feature map. The detailed description of our framework is elucidated as below.

In the stripe correction network, the first three layers are implemented as downsampling layers, using convolution (Conv), Instance Normalization (IN), and Rectified Linear Unit (ReLU). For input patches with dimensions of $256 \times 256 \times 3$, a convolutional layer is first applied to increase the feature depth to $256 \times 256 \times 64$. The feature map is then downsampled by two in the next two downsampling layers, while the number of channels is doubled by the convolution with stride 2. This process results in a $128 \times 128 \times 128$ feature map capable of extracting low-level visual features of the input patch. To extract high-level semantic features for describing visual

histological components, nine stacked residual blocks are employed. These blocks incorporate shortcut connections to facilitate the information flow during network training. The first two upsampling layers are realized by a transpose convolution with stride 2, followed by IN and ReLU. The final upsampling layer is realized by a convolutional layer with stride 1, followed by Hyperbolic Tangent (Tanh) activation. In addition, the discriminator of our model employs a relatively shallow CNN architecture, with the last convolution layer producing a single-channel feature map for a patch to be classified as either normal or anomaly.

For training SSCOR, we utilized the Adam solver¹⁴ with a learning rate of 0.0002 and exponential decay rates for the first and second moment estimates set to 0.5 and 0.999, respectively. During training, the number of epochs is set to 200, with a fixed learning rate in the first 100 epochs and a linear decay to zero over the next 100 epochs. The hyper-parameters of the sampling strategy are described in the Supplementary Table 6. During inference, for the local-to-global correction, we set the step size of sliding-window as half of the patch size or a fixed value of 100 pixels."

4. The network architecture here is somewhat simple, and attention mechanism and multi-scale mechanism are not involved. The image enhancement effect needed to be explained further.

Response: We thank the reviewer for this comment. To handle a sampled patch of 128×128 or 256×256 pixels (see Supplementary Table 6. Proximity sampling on representative cases) , which is relatively small, a network architecture with nine residual blocks is sufficiently deep and powerful. Therefore, additional attention or multi-scale techniques were not necessary, as they could make the network too redundant, leading to over-fitting that could degrade the correction effects.

In our model, the image enhancement effect is primarily achieved through our adversarial training scheme. Since SSCOR receives an anomaly patch and its nearby normal patch, they will be passed to the discriminator network to distinguish if they are normal. If the anomaly patch was not well corrected, the network can easily discriminate them. Hence, the objective is to enhance the quality of the anomaly patch to its maximum extent, thereby maximizing the difficulty of discrimination. The explanation of this principle is described in Method Section 5.2 (Network Architecture), as below.

"Adversarial losses. During training, an anomaly patch and its associate normal patch in

proximity will be passed to the discriminator network to distinguish if they are normal. If the anomaly patch was not well corrected, the network could easily discriminate them. Hence, the objective is to enhance the quality of the anomaly patch to its maximum extent, thereby maximizing the difficulty of discrimination. "

5. In the stage of strip correction and synthesis, if the artificial interferes of correction discipline, then there will be problems in the content consistency judgment of image blocks. How to ensure the accurate maintenance of image features?

Response: Thank you for bringing this to our attention. As stated in the Method Section 5.2 (Network Architecture), content consistency was achieved through the utilization of our content consistency loss (Equation 7), as described below. The content consistency loss involves two reciprocal processes, "destriping-shading" and "shading-destriping". Specifically, in the "destriping-shading" process, we first remove stripes from the input anomaly patch, and subsequently synthesize the stripes back to their corrected form, obtaining a synthesized patch. By constraining the synthesized patch to be similar to the input anomaly patch, we are able to maintain image content quality during network inference and achieve content consistency. Furthermore, we also implemented a reciprocal "shading-destriping" process on an input corrected patch to further reinforce content consistency. We have elaborated on this principle in detail in Section 5.2 Network Architecture.

"... to be jointly learned. The content consistency loss essentially introduces two reciprocal processes: "destriping-shading" and "shading-destriping". Specifically, in the "destriping-shading" process, we first remove stripes from the input anomaly patch, and subsequently synthesize the stripes back to their corrected form, obtaining a synthesized patch. By constraining the synthesized patch to be similar to the input anomaly patch, we are able to maintain image content quality during network inference and achieve content consistency. Furthermore, we also implemented a reciprocal "shading-destriping" process on an input corrected patch to further reinforce content consistency. Thus, the content consistency loss can be formulated as ..."

6. In equation 7, there should be symbol errors and missing.

Response: Thanks for bringing this to our attention. We have fixed the problem in the revision as below.

$$\mathcal{L}_{cons} = \mathbb{E}_X[\|G_S(G_C(X)) - X\|_1] + \mathbb{E}_Y[\|G_C(G_S(Y)) - Y\|_1]. \quad (7)$$

7. It is very important to evaluate the quality of image restoration. The application of Icv here should not be a very good choice.

Response: Thanks for your comments. In these literatures [1-4], the Inverse Coefficient Variation (ICV), or similar metrics, have been proven useful for evaluating the quality of image correction. Mean intensity fluctuations have been used to assess intensity profiles [1], while intensity standard deviations have been used to quantify the severity of stripes and shadowing artifacts [1, 2]. ICV, which combines mean intensity and the standard deviation, represents the ratio of mean signal intensity to the degree of intensity fluctuation in the stripe area. ICV has also been utilized to evaluate the stripe correction [3, 4]. A higher ICV value indicates more effective correction of stripes and artifacts. To our knowledge, ICV could be used as one of the metrics to evaluate the quality of stripe correction.

[1] Fahrbach FO, Simon P, Rohrbach A. Microscopy with self-reconstructing beams. *Nat Photonics* 4, 780-785 (2010).

[2] Glaser AK, et al. Multidirectional digital scanned light-sheet microscopy enables uniform fluorescence excitation and contrast-enhanced imaging. *Sci Rep* 8, 13878 (2018).

[3] Rakwatin P, Takeuchi W, Yasuoka Y. Stripe noise reduction in MODIS data by combining histogram matching with facet filter. *IEEE Transactions on Geoscience and Remote Sensing* 45, 1844-1856 (2007).

[4] Pollatou A. An automated method for removal of striping artifacts in fluorescent whole-slide microscopy. *J Neurosci Methods* 341, 108781 (2020).

In addition, to further evaluate the effectiveness of different correction methods, we have conducted an anonymous online survey with the participation of 50 researchers from diverse backgrounds. Please refer to Supplementary Note 3 for the details of the user study.

"Supplementary Note 3. User study

Subjective evaluation is a reliable method for assessing the quality of biomedical images, taking into account the human perception of the images. To further evaluate the effectiveness of different correction methods, an anonymous online survey was conducted with 50 participants from five different backgrounds, including computer vision, pathology, biophotonics, optics, and precision instruments. The survey included the test images of two types: (i) multiphoton microscopy (MPM) and fluorescence⁹ datasets with stripes, totalling 305 images and (ii) stimulated Raman scattering (SRS)¹⁰ datasets with synthetic stripes and artifacts, totalling 120 images. To fully compare the correction effect, the images for user study included not only the global images but also the local images. Each set of user study images included the input image

as well as the corresponding images corrected by the proposed SSCOR and the comparison methods (BaSiC¹¹, CIDRE¹², ZEN¹³, ZeroDCE², N2N⁷, and Mask-ShadowGAN⁵). The images were randomly assigned to the participants who were asked to rate a total of 26 sets of images on a severity scale from 0 to 10 according to the presence of stripes and artifacts in each image. The scoring criteria are as follows:

- (a) 0-2, worst, severe stripes and artifacts, significantly interfering with the observation.
- (b) 2-4, worse than average, heavy stripes and artifacts, interfering with the observation.
- (c) 4-6, average, mild stripes and artifacts, slightly interfering with the observation.
- (d) 6-8, better than average, slight stripes and artifacts, not affecting observation.
- (e) 8-10, best, almost no stripes and artifacts.

The image sets were randomly displayed, and the participants were allowed to return and change their previous rating until submission. The results of the user study were presented in Supplementary Table 1 and 2. The study showed that SSCOR significantly outperformed other methods for stripe correction and artifact removal. In addition, we visualized the user study results with regards to the background of the participants in Supplementary Fig. 9, implying the significant advantages of SSCOR in each discipline."

In the corresponding Results Section 2.4 (SSCOR removes special artifacts on stripes), we have made the following changes.

"We also showed the restoration results of real out-of-focus artifact, bubble artifact, and photobleaching artifact in the MPM images (Supplementary Fig. 8). In addition, to further evaluate the effectiveness of different correction methods, we conducted a user study involving 50 participants from diverse backgrounds (Supplementary Note 3). The results of the user study were presented in Supplementary Table 1 and 2, and Supplementary Fig. 9, which demonstrated that SSCOR significantly outperformed other methods for stripe correction and artifact removal in each discipline."

The results of user study demonstrated that SSCOR significantly outperformed other methods for stripe correction and artifact removal. Please see Supplementary Table 1 and 2.

Supplementary Table 1. User study results of MPM and fluorescence⁹ datasets with stripes. Fifty participants from diverse backgrounds were asked to rate the test images on a severity scale of 0 to 10 according to the presence of stripes and artifacts. Higher scores indicate better stripe correction. Results are presented as mean \pm SEM.

	Input	BaSiC¹¹	CIDRE¹²	ZEN¹³	SSCOR
Non-uniform	3.73 \pm 2.31	5.28 \pm 2.09	5.21 \pm 1.98	6.43 \pm 2.52	7.93 \pm 1.60
Grid	3.97 \pm 2.24	6.52 \pm 1.95	6.30 \pm 1.80	N/A	6.99 \pm 1.80
Oblique	4.98 \pm 2.17	5.96 \pm 1.90	5.67 \pm 1.90		8.11 \pm 1.41
Total	4.23 \pm 2.31	5.92 \pm 2.04	5.73 \pm 1.95	6.43 \pm 2.52	7.68 \pm 1.68

Supplementary Table 2. User study results of SRS¹⁰ datasets with synthetic stripes and artifacts. The comparison methods include ZeroDCE², BaSiC¹¹, N2N⁷, and Mask-ShadowGAN⁵. Results are presented as mean \pm SEM. Higher scores mean better stripe correction.

	Input	Other methods^{2,5,7,11}	SSCOR
Out-of-focus	2.88 \pm 2.21	4.89 \pm 1.77	8.57 \pm 1.15
Stripe	2.62 \pm 2.13	5.57 \pm 1.81	8.25 \pm 1.34
SFA	5.58 \pm 2.25	5.53 \pm 2.20	8.28 \pm 1.24
Bubble	2.57 \pm 2.16	5.64 \pm 1.97	7.91 \pm 1.24
Total	3.41 \pm 2.52	5.41 \pm 1.96	8.25 \pm 1.26

Additionally, we also added Supplementary Fig. 9 to visualize the user study results with regards to the backgrounds of participants, which showed the significant advantages of SSCOR in each discipline.

Supplementary Figure 9. User study results based on the background of participants. a SSCOR achieved superior performance on MPM and fluorescence⁹ datasets for different backgrounds (Nine participants in the pathology background are from the Affiliated Union Hospital and the First Affiliated Hospital of Fujian Medical University, respectively. Eleven participants in the biophotonics background are from Fujian Normal University. Ten participants in the optics background are from Xi'an Jiaotong University and Jilin University, respectively. Ten participants in the precision instruments background and ten participants in computer vision background are from Fuzhou University). **b** SSCOR outperformed other methods in correcting the SRS¹⁰ datasets with synthetic stripes and artifacts. Other methods include ZeroDCE², BaSiC¹¹, N2N⁷, and Mask-ShadowGAN⁵. The scores were shown as mean \pm SEM in box plots. Centerlines, medians; limits, 75% and 25%; whiskers, maximum and minimum. One-way analysis of variance (ANOVA) was utilized, followed by Tukey's multiple comparison test. The significance level is displayed as asterisks, and the value $P < 0.05$ is considered to be statistically significant (* $P < 0.05$, ** $P < 0.01$, *** $P < 0.001$, **** $P < 0.0001$; ns, not significant).

To illustrate the process of this user study, the representative pages of the questionnaire are shown below.

Image quality survey

Stitched fluorescence microscopy images inevitably have various types of stripes or artifacts, which seriously affect image quality and downstream applications. The purpose of this questionnaire is to compare the image correction ability of each method. Please read the following scoring criteria carefully to rate the image quality. There are 26 questions in total.

Stripes and artifact types are illustrated below

The scoring criteria are as follows

- 0-2, worst, severe stripes and artifacts, significantly interfering with the observation.
- 2-4, worse than average, heavy stripes and artifacts, interfering with the observation.
- 4-6, average, mild stripes and artifacts, slightly interfering with the observation.
- 6-8, better than average, slight stripes and artifacts, not affecting observation.
- 8-10, best, almost no stripes and artifacts.

* 01 Please enter your information.

For example: Precision instruments, Fuzhou University.

Please enter

* 02 Please rate the image quality after comparing the following images.

- 0-2, worst, severe stripes and artifacts, significantly interfering with the observation.
- 2-4, worse than average, heavy stripes and artifacts, interfering with the observation.
- 4-6, average, mild stripes and artifacts, slightly interfering with the observation.
- 6-8, better than average, slight stripes and artifacts, not affecting observation.
- 8-10, best, almost no stripes and artifacts.

Next page

Figure. Representative pages of the survey questionnaire.

8. In short, it is very difficult to realize automatic correction of microscopic image stripes so far, because it involves different imaging conditions and equipment as well as parameters selection and amplification resolution, etc. Second, there are some problems in the network structure design, and the handling of complex situations should be further explained. The selection of normal image blocks and image quality evaluation are very important and should be further studied.

Response: We would like to express our gratitude and acknowledge that your insightful comments have greatly contributed to improving the quality of our manuscript. Fully automated correction of microscopic image stripes will indeed bring convenience to users. Despite the fact that state-of-the-art correction methods, such as BaSiC or CIDRE, are capable of achieving automated uniform stripe correction, these methods require prior knowledge of the tile size, tile overlap ratio, and other information related to the raw data.

In contrast, our method can correct non-uniform, oblique, and grid stripes, as well as removing scanning, bubble, and out-of-focus artifacts, requiring small amount of human interaction only. At this stage, we did not design SSCOR as a fully automated model because our method aims to address a wider range of tasks, including complex stripes and artifacts. A fully automated strategy may limit the range of applications for SSCOR. Besides, through the comprehensive experiments on four datasets of different modalities, the prior-free SSCOR is able to overcome the effects of different imaging conditions and equipment as well as parameters selection and amplification resolution.

In summary, we wholeheartedly acknowledge that a proper sampling strategy serves as a critical prerequisite for optimizing the efficacy of stripe correction. To this end, we have added three Supplementary Materials that provide detailed information on the different types of stripes and their corresponding sampling strategies. Additionally, we have further elaborated on the underlying network structure, and conducted a user study to assess the effectiveness of various correction methods. Thanks to your invaluable feedback, we intend to incorporate the development of a fully automated SSCOR into our forthcoming research program.

Reviewer #2:

The manuscript entitled, "A deep learning-based stripe self-correction method for stitched microscopic images", the authors develop a method to remove the stripping artifact and other anomalies from large, tiled images. The use of paired normal an

anomaly patches to reciprocally train the ablation studies on the effectiveness of sampling strategy, adversarial reciprocal training, and local-to-global correction are a strength of the manuscript. Considering the increasing dependence on imaging and image analysis in biological research this is timely and important. Tools to improve the accuracy and reproducibility of imaging data analysis will be beneficial across multiple disciplines. Overall, the manuscript is clear and logical with strong validation of the methods. There are only a few concerns that should be addressed before publication.

Response: We sincerely appreciate your gracious feedback and wholeheartedly agree that continued efforts in this direction will yield significant benefits across multiple disciplines. We have diligently addressed the concerns you have raised, as elaborated below. The corresponding changes to the resubmitted manuscript have been highlighted in red.

1. The images chosen for the figures are relatively even in GT fluorescence signal across the tissue. It is unclear if at least one channel of fluorescence needs to be fairly homogenous across the tissue/sample for effective correction. To demonstrate that SSCOR does not over or under correct it would be good to test on an image that has GT heterogeneity in fluorescent signal. This is quite important for analysis where the intensity of a particular marker is a comparison of low vs high expression. Further, would the location of a strip and subsequent correct impact the accuracy of heterogenous fluorescence intensity? For example, does strip correction in a region of low-intensity lead to overestimated signal or conversely does strip correction in a region of high fluorescence intensity underestimated the signal?

Response: We appreciate the comments of the reviewer. We would like to emphasize that our proposed method does not assume homogeneity across the tissue sample. In practice, from the perspective of human vision, stripes are more noticeable in weak signal images with homogeneous tissue. For instance, for label-free multiphoton images, we would like to highlight that the second harmonic generation (SHG) signal, which is color-coded green in our images, primarily arises from mesh-like collagen fibers. Conversely, the two-photon excited fluorescence (TPEF) signal, which is color-coded red, is mainly derived from relatively homogeneous components, such as nicotinamide adenine dinucleotide (NADH) and flavin adenine dinucleotide (FAD). Therefore, the stripes tend to be more noticeable in homogeneous TPEF images with weak signals, while they are relatively less obvious in heterogeneous SHG images with strong signals. Actually, stripes exist in both the SHG and TPEF channels of

multiphoton images.

In order to demonstrate that our method does not result in the accuracy of fluorescence intensity in heterogeneous images, we have analyzed the intensity profiles of the stripe and non-stripe regions in a MPM image (Supplementary Fig. 4a) with two heterogeneous channels separately. Additionally, we have synthesized stripes on a relatively homogeneous SRS image, and showed our method does not overestimate the fluorescence signal in low-intensity regions or underestimate that in high-intensity regions. These results demonstrate that our proposed correction method preserves the fluorescence intensity contrast of tissue components without altering the fluorescence intensity of either the stripe or non-stripe regions in the entire image. Please see Supplementary Fig. 4 as below.

Supplementary Figure 4. Unbiased signal estimation across intensity levels of heterogeneous images. a The intensity profiles of the stripe and non-stripe regions in the MPM image with real stripes. SSCOR is capable of correcting stripes and preserving the fluorescence

intensity contrast in non-stripe regions in two heterogeneous channels. **b** Comparison of correction effects in the heterogeneous SRS image with synthetic stripes. **c** and **d** The enlarged high-intensity region (red boxes) and low-intensity region (yellow boxes) in **b**. The intensity profiles demonstrated that SSCOR does not overestimate or underestimate fluorescence signal in stripe regions, while still preserving the intensity contrast of tissue components in non-stripe regions.

We have accordingly revised Results Section 2.2 (SSCOR adaptively corrects different types of stripes), as below.

"For more details, the intensity profiles of the non-uniform stripes and grid stripes, as well as the other typical stripe correction examples were shown in Supplementary Fig. 3. Besides, we illustrated the correction results across different intensity levels of heterogeneous images in Supplementary Fig. 4, which demonstrate that SSCOR does not overestimate or underestimate the fluorescence signal in stripe regions while still preserving the intensity contrast of tissue components in non-stripe regions."

2. Minor comment - When discussing collagen quantification, the analysis software (CT-Fire) quantifies the number and density of collagen fibers, not total collagen. The word "fiber" should be included in the text.

Response: We appreciate the reviewer's comment. We have promptly addressed the issue, as per the reviewer's suggestion, and made the necessary corrections. In the revised manuscript, we have updated the Results Section 2.5 (SSCOR-corrected images benefit downstream tasks).

"In the Fig. 5d, we extracted three prognosis-related signatures including the collagen fiber number³¹, collagen fiber density³¹, and contrast feature of collagen fiber³² in the SRS image with out-of-focus artifact. In the visualization of collagen fiber number and density, ZeroDCE seemed to remove artifacts area, but it also produced incorrect image content."

The corresponding revised Fig. 5 is shown as below.

Fig. 5 SSCOR-corrected images benefit downstream tasks. **a** Schematic diagram of SSCOR applications in three downstream tasks. Virtual staining²⁷, cell counting³⁰, and collagen fiber extraction^{31,32} are performed to verify the significance of SSCOR in typical stripe and artifacts correction ... **e** SSCOR-corrected image restores the tumor collagen-associated signatures more realistically, which is reflected in the visualization of collagen fiber number and density. The green lines indicate the orientation of collagen fiber. The red dots indicate the location of collagen fiber. The heatmap is used to visualize the collagen fiber region. The SSCOR-corrected image is also highly consistent with GT image in quantitative comparison of collagen fiber density, number, and contrast. Scale bars: 100 μm .

3. Figure 4D. It is unclear what the 4 different graphs represent. Does each represent a different field of view or region of interest within the tiled image?

Response: Thank you for bringing this to our attention. Fig. 4d provided PSNR and SSIM of the images in Fig. 4a. We have revised Fig. 4d accordingly and clarified the caption to avoid any potential confusion.

Fig. 4 The correction results of special artifacts on stripes. ... **d** Compared with other methods, SSCOR-corrected images have better image quality, as indicated by PSNR and SSIM of the images in **a**.

Reviewer #3:

Summary:

This paper presents an unsupervised patch-wise deep learning method for correcting stripes and other artifacts for stitched microscopic images. The deep learning-based correction is performed in a patch-wise manner, and the whole image can be processed in a sliding window manner. The authors propose the proximity sampling strategy to sample normal and abnormal patches in the given image and adopt the unsupervised CycleGAN strategy to train the model. The authors conduct experiments on three datasets and demonstrate the effectiveness of the proposed approach on stripe corrections and artifact removal.

Strengths:

1. The paper provides a solution for unsupervised stripe correction and artifact removal, which is an important problem in microscopic image processing, considering the difficulty of acquiring paired/annotated datasets.

2. The paper is well written, and the authors present their approaches and results clearly.

3. The authors validate the benefits of SSCOR for various downstream tasks.

Response: I would like to express my sincere gratitude for your positive feedback regarding our manuscript. Your comments have been very encouraging, and we appreciate your recognition of the importance of our study. As you noted, our research proposes an unsupervised strategy to address critical challenges in the field of microscopic imaging, specifically stripe correction and artifact removal. We firmly believe that this approach can have a significant impact on multiple disciplines, and we are excited about its potential.

We have taken your comments into careful consideration, and have made revisions to our manuscript accordingly. We have provided a point-by-point explanation of these changes below. The corresponding changes to the resubmitted manuscript have been highlighted in red.

Weakness:

1. The impact/novelty is limited. Deep learning-based stripe/artifacts correction is not a new topic. As mentioned in the paper, there are some supervised learning-based approaches for that. One of the key points is the unsupervised learning manner. While unsupervised learning is achieved with the well-studied CycleGAN strategy. From the technical aspect, the novelty is incremental.

Response: We appreciate the comments of the reviewer. As mentioned, from 2014 to 2017, three significant image processing-based stripe correction methods were developed to address this issue (Nature Methods 11, 602 (2014) [1], Nature Methods 12, 404-406 (2015) [2], and Nature Communications 8, 14836 (2017) [3]). Existing methods focus on uniform stripe correction, requiring the prior knowledge of the tile size or raw stitching data. This can be challenging to meet the requirements of complex or multi-type stripe correction in practical imaging experiments.

Inspired by this, our approach offers a novel and unique perspective on this problem. Firstly, instead of tile-based sampling scheme, we adopt an innovative patch-based sampling scheme for obtaining a large number of samples demanded by deep model training. Specifically, we decompose high-resolution images into small normal patches without shadings and anomaly patches with various shadings. In this way, we can collect sufficient and diverse training data without extra large efforts. Since the normal and abnormal patches are unpaired, we can employ a GAN-based deep learning model to learn their mappings in an unsupervised manner.

Another novelty of our approach is the proposed proximity sampling strategy, which enables the self-training scheme to effectively benefit from the collected normal and abnormal patches from the respective regions. In practice, randomly sampling normal and abnormal patches leads to sub-optimal results, as shown in the ablation study on sampling strategy in Supplementary Note 5 and Supplementary Fig. 16. This is because normal and abnormal patches, which are located far apart, usually have significantly different textures and intensities, making it difficult to learn the mappings between them. Our proposed proximity sampling method adds a simple yet effective contextual constraint to the sampling process. It forces the embeddings of sampled normal and abnormal patches to be close.

The reason why we chose a generic image enhancement framework for our approach is that our solution aims at a wide range of stripes and artifacts, unlike other complex deep learning models that may introduce inductive bias to handle specific types of stripes or artifacts. By

using a generic framework with little inductive bias, we can achieve robust performance across various types of stripes and artifacts without significant degradation.

[1] Coster, A. D., Wichaidit, C., Rajaram, S., Altschuler, S. J. & Wu, L. F. A simple image correction method for high-throughput microscopy. *Nature Methods* 11, 602-602 (2014).

[2] Smith, K. et al. CIDRE: an illumination-correction method for optical microscopy. *Nature Methods* 12, 404-406 (2015).

[3] Peng, T. et al. A BaSiC tool for background and shading correction of optical microscopy images. *Nature Communications* 8, 14836 (2017).

2. The motivation/description/rationale of some key components in the methodology part is unclear.

a) The proposed approach adopts patch-wise correction and uses the "local-to-global" strategy to correct the whole image. However, the authors should clarify how they ensure the consistency/smoothness of different patches when merging the patch-based results. Otherwise, this manner may introduce secondary stripes in the image.

Response: Thank you for bringing up this point. To correct an input stitched image, we employ a sliding-window approach to partition it into multiple overlapping patches. These patches are then individually processed through a previously well-trained stripe correction network. Following correction, the patches are merged by averaging the overlapped corrected image regions over the adjacent sliding windows, ultimately reconstructing a complete image with the same resolution as the original stitched image. The rationale behind implementing a local-to-global strategy is to enable effective correction of the same pixel positions across different patches, and to incorporate more contextual information.

In practice, this approach facilitates smoother texture alignment between corrected patches and mitigates the issue of secondary stripes. Its effectiveness is demonstrated in the ablation study presented in Supplementary Fig. 16. As per the reviewer's comment, we have elucidated the process of the "local-to-global" strategy in Supplementary Note 5 as below, and added Supplementary Fig. 2 to better describe the procedure.

"Supplementary Note 5: Ablation study

Local-to-global Correction

As the final stage of SSCOR workflow, local-to-global correction consists of two steps, i.e.,

local correction and global merging. First, the whole-slide images are partitioned into overlapping local patches in the sliding-window manner and fed into the stripe correction network to obtain corrected patches, which is the local correction step. Next, all the corrected patches are merged by computing the average of overlapped region between adjacent patches. The merging procedure yields a whole image with the same resolution as the original stitched image, which is the global merging step. This integration of local and global steps enables the suppression of stripes and fosters a smoother texture alignment between corrected patches."

Supplementary Figure 2. Illustration of the local-to-global strategy. The local-to-global strategy consists of the local correction and global merging steps. The local correction step involves partitioning the input stitched image into overlapping local patches using a sliding-window approach. Each local patch is then processed by the well-trained stripe correction network. In the global merging step, the corrected patches are merged and the overlapped region of adjacent local corrected patches are averaged to reconstruct the complete image. In practice, the sliding-window step size is set to be smaller than the patch size, to ensure that adjacent patches overlap sufficiently to enable the smoothness of merging.

b) The proposed approach heavily relies on the sampled normal/abnormal patches, while this sampling part involves too much human intervention from the description in the supplementary material. For example, it needs humans to manually annotate the partition of tiles, determine the dimension of patches, and also the manner to get the normal patches (with human-defined rules). As we need to conduct this sampling/training procedure for each dataset (or image), how do you ensure the robustness and generalization of the proposed framework for new testing cases? Especially for the sampling of normal patches in the images with artifacts, the adopted strategy is quite objective. Can you quantitatively describe the adopted strategy?

Response: Thank you for your comment. Firstly, our proposed proximity sampling strategy actually requires minimal human intervention and does not rely on precise estimation of tile sizes like some tile-based methods such as CIDRE or BaSiC. As described in Section 2.3 (Fig. 3), our experiments have shown that SSCOR is more tolerant and robust against stripes with cropping or rotation, indicating that even with imprecise sampling of normal/abnormal patches on and off stripes, our model can still correct the stripes effectively. In the revision, to avoid the confusion, we have added a Supplementary Fig. 6 as shown below, which demonstrates that SSCOR has tolerance towards the imprecise user-defined abnormal region for bubble-like artifacts. As a result, Fig. 3 and Supplementary Fig. 6 jointly demonstrated the robustness of the proposed SSCOR in a quantitative manner. Accordingly, we updated Results Section 2.3 (High tolerance towards imprecise prior stitched information) as below.

"Besides, SSCOR also showed tolerance towards the imprecise user-defined abnormal region for bubble-like artifacts (Supplementary Fig. 6). To sum up, SSCOR not only requires little raw information to correct stripes, but also demonstrates high tolerance and robustness against imprecise prior image adjustments."

Supplementary Figure 6. SSCOR has tolerance towards the imprecise user-defined abnormal region for bubble-like artifacts. **a** We illustrate the user-defined abnormal region for sampling anomaly patches from bubble-like artifacts. As shown in the schematic diagram, the user-defined boundary can be roughly adjusted to zoom in or out by 10%. **b** and **c** The yellow boxes represent the user-defined abnormal regions for bubble-like artifacts with different errors. We present the correction results obtained by sampling from different abnormal regions, with the enlarged and highlighted local regions within red boxes. Furthermore, we measure the corresponding PSNR and SSIM. Our observations reveal that SSCOR effectively corrects artifacts, even when the boundary error ranges from -20% to 20%. Note that the superior artifact correction can be achieved by sampling anomaly patches from an abnormal region that is equal to or larger in size than the artifact itself. This suggests that the abnormal region can be slightly larger than the artifact to attain a high-quality correction outcome. Scale bar of input image: 1 mm. Scale bar of enlarged image: 250 μm .

Secondly, the generalizability of SSCOR can be demonstrated by the experimental results on three image datasets, achieving the state-of-the-art performance across different imaging conditions and modalities. To further show the generalizability of SSCOR, we have conducted tests on two additional publicly available H&E datasets, CoNSep and MoNuSAC (please refer to Supplementary Table 4).

Supplementary Table 4. Comparison of the microscopic datasets.

Stripes/Artifacts types	Dataset	Specimen	Number	Resolution (pixel)	Microscope	Detector /Sensor	Objective	Immersi on medium	Light Source		
									Type	Detail	
Stripes	H&E ^{15,16}	Colorectal adenocarcinoma	14	1000×1000	Omnyx VL120	-	40×	-	LED	-	
		Breast cancer, kidney cancer, lung cancer, and prostate cancer	3	1600×1400 1800×1400 2000×1400	-	-	40×	-	LED	-	
		Breast cancer	1	11143×9249	Motic VM1000	CCD	40× 0.95 NA	Water	LED	Electronically dimmable 6V/10W	
	MPM	Breast cancer	10	3430×3430,	Zeiss LSM 880	PMT	20× 0.8 NA	Air	Coherent Laser	Chameleon Ultra	
		Cerebral vascular malformation	1	5550×5550,						Ti: Sapphire Femtosecond Excitation wavelength (810 nm)	
Grid		Liver cancer	4	4403×4403,					Average laser power (30 mW)		
Oblique	Fluorescence ⁹	Mouse brain	5	4171×3736, 4346×3080,	Hamamatsu NanoZoomer	TDI	20× 0.75 NA	Air	Hamamatsu Mercury lamp	LX2000 Ultrahigh-pressure 200 W	
Artifacts	Out-of-focus			7350×5390,						picoEmerald Tunable Two-Color Source	
	Bubble	SRS ¹⁰	Glioblastoma	2		Olympus FV300	CCD	60× 1.2 NA	Water	APE GmbH Laser	Picosecond Optimal Raman shifts (2973, 2921, and 2851 cm ⁻¹)
	SFA			2838×3892,							

H&E, Haemotoxylin and Eosin; MPM, multiphoton microscopy; PMT, photomultiplier; NA, numerical aperture; TDI, time delay integration; SRS, Stimulated Raman scattering; CCD, charge coupled device.

We have also revised Results Section 2.5 (SSCOR-corrected images benefit downstream tasks), and added the representative correction results in Supplementary Fig. 11, see below.

"The correction results of other representative synthetic H&E stripe images are presented in Supplementary Fig. 11. As a conclusion, SSCOR is in favor of not only recovering the ambiguous segmentation of cells, but also improving cell classification. "

Supplementary Figure 11. Representative correction results of synthetic stripes on H&E images. We synthesized non-uniform stripes on the H&E images from two public datasets, CoNSep¹⁵ and MoNuSAC¹⁶. Comparing to BaSiC, SSCOR obtains better stripe correction results.

Finally, to provide a more comprehensive understanding of our proposed proximity sampling strategy, we have included a user guidance that provides step-by-step correction assistance, including a schematic diagram of typical stripes and artifacts (Supplementary Fig. 15), along with the corresponding description and sampling strategy procedure (Supplementary Table 5). Additionally, we have included the representative sampling cases that were experimented (Supplementary Table 6).

In the revision, we have revised the Method Section 5.3 (Proximity Sampling Strategy) as below, which also adds a reference to Supplementary Fig. 15 and Supplementary Table 5.

"Proximity sampling strategy serves as one of the most critical components in SSCOR. For any stitched images with different stripes and artifacts, the sampling strategy can be summarized as: 1) define normal and abnormal regions; 2) iteratively sample an anomaly patch from abnormal region and a normal patches from normal region in proximity to pair with the normal patch. Specifically, ...

... To improve the usability of SSCOR for correcting various types of stripes or artifacts, we drew the schematic diagrams for illustrating six typical stripes and artifacts (Supplementary Fig. 15). Additionally, a comprehensive description of the various types of stripes and artifacts,

along with their corresponding sampling strategies, is provided in Supplementary Table 5. Detailed settings of the proximity sampling strategy applied to the experimented images are outlined in Supplementary Table 6. ..."

Supplementary Figure 15. Schematic diagram of typical stripes and artifacts. We show the diagrams of non-uniform stripe, grid stripe, oblique stripe, bubble-like artifact, scanning fringe artifact, and out-of-focus artifact. The sampling strategy of the stripes and artifacts are outlined in Supplementary Table 5.

Supplementary Table 5. Description of stripe and artifact types and the corresponding sampling strategy.

Type	Description	Causes	Sampling strategy
Non-uniform stripe	The stitched image presents non-uniform stripe pattern, in which each tile exhibits diverse shading patterns.	Multiple factors like the condition of specimen or microscope result in non-uniformity.	Since the exact positions of stripes in stitched images are unknown, we assume that the stripe are uniformly distributed and perpendicular to the image boundaries. This assumption applies even to images with oblique stripes. The sampling steps are as follows:
Grid stripe	The stitched image presents horizontal and vertical stripes that resemble a grid pattern, in which each tile shares the same shading pattern.	The shading of each tile is caused by uneven illumination.	 1. Estimate the position of each stripe in the stitched image based on image width, height, and the number of stripes. 2. Sample anomaly patches along each estimated horizontal or vertical stripe using a sliding window approach.

Oblique stripe	The vertical or horizontal stripes of stitched images present slightly oblique.	Pathological tissues are commonly adhered askew on glass slide for microscopic examination, which may necessitate proper rotation of the acquired image to meet downstream application requirement. This rotation can lead to the presence of oblique stripes in the image.	3. The striped area covered by the sampling is considered the abnormal region, with the remaining non-striped area being the normal region. 4. Once an anomaly patch is sampled, the corresponding normal patch can be obtained from the nearby non-striped areas using the proposed strategy (please refer to Sec. 5.3 in Method).
Bubble-like artifact	The artifact appears bubble-like irregular shape with uneven shading.	The artifact is essentially local tissue moisture loss caused by laser photodamage or the unsealed specimen.	1. Localize the abnormal region through user interaction, where the user can draw a coarse region around the artifacts to define the abnormal region. The remaining image region can be considered as normal region. 2. Sample anomaly patches within the abnormal region, while normal patches can be sampled from adjacent normal regions outside the abnormal region.
Scanning fringe artifact	The artifact typically appears throughout the entire image, accompanied by a large area of noise.	The artifact is often produced by a high-speed galvo-resonant scanning imaging system.	
Out-of-focus artifact	The artifact usually locates in the corner of the stitched image, which appears significantly reduction of the signal intensity within shading.	Due to the unevenness of the boundary area of the tissue sample, the microscope is out of focus on these areas.	
Co-existed artifacts and stripes	The aforementioned artifacts overlap with the stripes in the stitched image.	Refer to the above reasons with regards to the type of stripe and artifact.	
			1. Apply the aforementioned strategies jointly to define abnormal regions encompassing both stripes and artifacts. 2. Sample anomaly patches from abnormal regions, while normal patches are sampled from nearby normal regions.

Supplementary Table 6. Proximity sampling on representative cases.

Stripes/Artifacts types	Specimen	Resolution(pixel)	Reference	Tiles	Sample strategy		
					Step size (pixel)	Patch size (pixel)	Sample number
Stripes	Cerebral vascular malformation	2881×2872	Supplementary Table 3	3×3	64	256×256	166
		6387×6387	Supplementary Fig. 7b	7×7	128		576
	Non-uniform Breast cancer	3430×3430	Supplementary Fig. 3a	7×7	128	256×256	198
		6348×5376	Fig. 2a	11×13	128		633
		11109×9122	Fig. 5c	5×5	256	512×512	316

			4172×3737	Fig. 3b	9×10	64		210
	Oblique	Mouse brain	4354×3086	Fig. 2a	8×11	64	256×256	220
			5000×3750	Supplementary Fig. 3c	11×13	128		231
	Grid	Liver cancer	4403×4403	Fig. 2a	9×9	128	128×128	544
	Out-of-focus		7350×5390	Fig. 4	11×15	128		735
Artifacts	Bubble	Glioblastoma	7350×5390	Fig. 4	11×15	128	256×256	735
	SFA		7350×5390	Fig. 4	11×15	128		470

c) One of the key points for training deep models is avoiding underfitting or overfitting. If the model is underfitting, it cannot correct the strips well. If the model is overfitting, it maybe generates some over-smooth results and discards some important content details. This point is more important in the proposed approach, as we need to train one model for each testing data. Unfortunately, the authors did not describe/discuss this point in detail.

Response: Thank you for bringing this point to our attention. To achieve the optimal performance while avoiding underfitting and overfitting, based on our experience, most hyper-parameters of training SSCOR were set to the same values for different images. The only difference lies in the step size of sliding-window sampling, which is proportional to the resolution of the images and thus determines the number of training sample patches. Following the suggestion of the reviewer, we have added more training details in Supplementary Note 4.

"Supplementary Note 4: Network implementation details

... For training SSCOR, we utilized the Adam solver¹⁴ with a learning rate of 0.0002 and exponential decay rates for the first and second moment estimates set to 0.5 and 0.999, respectively. During training, the number of epochs is set to 200, with a fixed learning rate in the first 100 epochs and a linear decay to zero over the next 100 epochs. The hyper-parameters of the sampling strategy are described in the Supplementary Table 6. During inference, for the local-to-global correction, we set the step size of sliding-window as half of the patch size or a fixed value of 100 pixels."

Moreover, we have outlined the sampling strategy settings used for the representative images in our paper in Supplementary Table 6 (Please refer to the response to the previous question). As observed in Supplementary Table 6, we adopted a step size of 128 for sampling in most cases. For relatively smaller images, such as the cerebral vascular malformation specimen with a resolution of 2881×2872 pixel, we set the step size to 64 for collecting enough training patches. In contrast, for larger images, such as the H&E image with a resolution of 11109×9122 pixel, we set the step size to 256 for saving training time. Additionally, we have quantitatively analyzed the effects of step size on model performance in the figure below, indicating that our model is stable and the step size generally does not significantly affect the model performance.

Figure. The effect of sampling step size on model performance. **a** and **b** We adopted different step sizes during training, and SSCOR showed the stable performance with the step sizes of 64, 128, or 256. Despite a slight performance drop at the step size of 512, SSCOR is able to well correct stripes. The white arrow indicates the partially uncorrected stripes when the step size is 512. Scale bar of **a**: 1 mm. Scale bar of **b**: 500 μ m. **c** The PSNR and SSIM curves for the corrected images exhibit minimal fluctuations, indicating the stability of SSCOR.

d) In the discussion part, the authors mentioned that previous methods like CIDRE, BaSiC and ZEN may achieve slightly better performance than SSCOR when knowing the prior information on tile partition. In the proposed framework, we also need to roughly annotate the portion of tiles. In this case, what is the advantage of the proposed approach?

Response: Thank you for your comment. Notably, precise tile size is usually obtained from raw data, which is difficult to estimate during practical correction process. In this situation, CIDRE and BaSiC need precise tile size as prior, and Zen requires the raw image to complete the correction. Moreover, these methods have struggled with addressing non-uniform stripes and artifacts, despite having prior knowledge.

In comparison, our approach effectively eliminates the requirement of precise tile size, as evidenced in Fig. 3. Besides, our approach offers distinct advantage over CIDRE, BaSiC, and ZEN due to its adaptive approach to stripe correction, which can handle non-uniform, oblique, and grid stripes, as well as remove scanning, bubble, and out-of-focus artifacts. These advantages are expected to have a significant impact on downstream applications in multiple fields. Table 1 provides a comprehensive summary of the capabilities, advantages, and limitations of SSCOR compared to the comparison methods.

In the revised manuscript, we have further emphasized the difference between our method and the previous method in terms of the requirement for tile partition. Please refer to Table 1.

Table 1 Comparison of the capabilities of different methods

Method	Ability	Applicable Scenarios	Limitations	Correction Time (s)	
				Image Size (pixel): 6348×5376 (13×11 tiles) Tile Size (pixel): 512×512	Image Size (pixel): 2881×2872 (3×3 tiles) Tile Size (pixel): 1024×1024
ZEN ¹⁵		ZEN can correct uniform stripes in the raw image acquired by Zeiss microscope before online stitching	ZEN demands over 300 tiles per image for satisfactory correction performance, but over-large raw files may cause ZEN software overload and crash	3.7	3
BaSiC ¹⁴	Uniform stripe correction	BaSiC and CIDRE can handle a stitched image given the information of each tile under uniform shading condition	BaSiC and CIDRE are designed to process the raw stitching tiles. Once the image has been stitched or cropped, these methods should estimate the tile size first, manually crop tiles, and then correct the uniform shading of each tile. They can hardly handle uniform stripes.	57.43	4.49
CIDRE ¹¹				55.95	13.5
N2N ²³	Denosing	N2N can denoise the Gaussian noise and Poisson noise in synthetic fluorescence images, as well as the natural noise distribution in real-world noisy images	Since the noise pattern of scanning fringe artifacts (SFA) is significantly different from natural noises, N2N can hardly recover the original tissue signal from SFA in microscopic images	182.63	49.5
ZeroDCE ²²	Out-of-focus artifact removal	ZeroDCE can recover out-of-focus-like low-light images, and reduce the influence of the lighting environment on the object color under inadequate lighting conditions	ZeroDCE is designed to enhance the low-light natural images, while it is not perfectly suitable for restoring the microscopic images	288.79	59.03
Mask-ShadowGAN ²⁴	Bubble-like artifact removal	Mask-ShadowGAN can properly remove the bubble-like hard shadows caused by objects blocking and preserve the texture details	Mask-ShadowGAN uses binary masks to represent shadow regions, so it is better at handling hard (uniform) shadow, but less effective at handling soft (non-uniform) shadow	234.3	53.89
SSCOR	Various stripe correction and artifact removal	SSCOR can adaptively correct non-uniform, oblique, and grid stripes, as well as remove scanning, bubble, and out-of-focus artifacts, while faithfully preserving the original image content	At present, SSCOR can only perform near-real time correction. When the image stripe is uniform and the precise tile size is known, BaSiC and ZEN may show better correction effect.	189.49	45.26

The correction time of Zen¹⁵, BaSiC¹⁴, and CIDRE¹¹ were tested by the CPU platform (Intel Core i5). The correction time of SSCOR, Neighbor2Neighbor²³, ZeroDCE²², Mask-ShadowGAN²⁴ were tested by the GPU platform (NVIDIA RTX 3090). The training requirements for images were detailed in Supplementary Table 3.

In addition, we have updated the Discussion Section as shown below.

"When handling the uniform stripes, with knowing the prior information on **precise** tile partition, previous methods like CIDRE, BaSiC, and ZEN can achieve slightly better performance than SSCOR."

3. For the presented results (Figs.2&3&), there are still many stripes in the corrected image, although the proposed approach achieves better results than other baseline methods. This may indicate that the proposed approach may have limitations in correcting heavy stripes.

Response: Thanks for pointing this out. As shown in Fig. 3, our proposed method, SSCOR, has high tolerance and robustness when it comes to imprecise adjustments made to prior images, which highlights the broad range of errors that our method can effectively correct. Specifically, our approach is capable of maintaining stable correction as long as the cropping error is within 16% of the tile size, or if the angle of the oblique stripes is less than 7 degrees, as demonstrated in Results Section 2.3.

It is acknowledged that the perfect correction for heavy stripes may not be achievable using our method. This is due to the fact that heavy stripes often result in a significant loss or degradation of original image content through shading. The stripe correction model may have difficulty distinguishing between the heavy stripes and the original low-intensity content in the image, potentially resulting in incorrect corrections. In these cases, the model may either hallucinate erroneous details to compensate the degraded image content caused by heavy stripes, or choose to neglect the stripes due to treating them as part of the original image content. In the revised manuscript, we have added the limitation of SSCOR for heavy stripes in the last paragraph of Discussion, as below.

"**Second, SSCOR may have difficulty distinguishing between the heavy stripes and the original low-intensity content in the image, potentially resulting in incorrect corrections. In these cases, the model may either hallucinate erroneous details to compensate the degraded image content caused by heavy stripes, or choose to neglect the stripes due to treating them as part of the original image content.** Third, both the adversarial self-training stage and local-to-global correction stage of SSCOR require GPU-based computation resources and cost more running time than previous methods."

Other comments:

1. Please clarify the robustness of the proposed proximity sampling strategy. Can it "correctly" sample normal regions under different situations? For example, there are artifacts near the stripes.

Response: Thanks for raising this issue. It is worth noting that stripes and artifacts often coexist, as demonstrated in Fig. 4 and Supplementary Fig. 6. In the revision, to provide a clearer understanding of the robustness of our proposed proximity sampling strategy, we have added a guidance to provide users with correction assistance, including a schematic diagram of typical stripes and artifacts (Supplementary Fig. 15), along with the corresponding description and sampling strategy (Supplementary Table 5). The supplements from Supplementary Fig. 1 and 15 as well as Supplementary Table 5 have improved the implementation details on how to sample normal patches under different situations. Please refer to the response of Question 2(b) above.

2. Please discuss more details about the training procedure, including whether it is difficult to train the GAN-based framework, the stability of the training procedure, and whether any pretrained weights are used in the model.

Response: Thanks for the suggestion. We did not utilize any pre-trained weights during the training process. In practice, we trained our model using the hyper-parameters as described in Supplementary Note 4 (Please refer to the response of Question 2(c) above). In addition, we have also examined the stability of the training process by randomly permuting the initial weights of the model, yielding similar final results (see the Figure below). Upon acceptance, we will release our training code, so as to facilitate reproducing our results.

Figure. Model performance versus the training epochs with random initialization. We show the training curves with regards to the correction quality for the synthesized stripes and out-of-focus artifacts, respectively. For each synthesized image, model training is performed

for three times with random initialization, which was measured in PSNR and SSIM for every 10 epochs until it reaches 200 epochs. As observed, both PSNR and SSIM exhibit an increasing trend with more training epochs. Although three training curves show fluctuations during training process, they eventually converge to similar results at the 200th epoch.

3. In Fig. 5c, please show the correction results of BaSiC and SSCOR. For the enlarged patches, it seems that the correction effect for these two patches is mild. I am not sure why this correction can improve the segmentation/classification performance.

Response: Thanks for your comments. Subtle alterations, such as the presence of noise in stripes or artifacts, have the potential to disrupt the deep visual features utilized for segmentation, subsequently resulting in the misidentification of cell boundaries and nuclear-cytoplasmic ratio by the segmentation model. According to the reviewer's comment, we have added more results to show the improvement.

Firstly, to compensate the results in Fig. 5c of the manuscript, we have shown the correction results of BaSiC and SSCOR in Supplementary Fig. 10a, and highlighted the cell classification results of another stripe-free local region different from the one shown in Fig. 5c. Based on the examination of experts, the SSCOR-corrected image effectively mitigates the adverse effects of stripes on the deep-learned cell classification model.

Secondly, in Supplementary Fig. 10b and 10c, we have shown the corrected synthetic stripe image from the publicly available H&E segmentation/classification dataset, CoNSep. The segmentation and classification performance of the corrected images were quantitatively evaluated using ground-truth annotations as the reference standard. We have also highlighted the improvement by comparing the result of the uncorrected local region with the SSCOR-corrected region.

We have incorporated the above results and updated the Results Section 2.5 (SSCOR-corrected images benefit downstream tasks), as below.

"In Supplementary Fig. 10a, we showed the complete corrected source image of Fig. 5c and highlighted the cell classification result of another stripe-free region. In addition, the classification results of a synthetic stripe image from another H&E dataset, CoNSep²⁹, were demonstrated and quantitatively evaluated in Supplementary Fig. 10b and 10c. The correction results of other representative synthetic H&E stripe images are presented in Supplementary Fig.

11. As a conclusion, SSCOR is in favor of not only recovering the ambiguous segmentation of cells, but also improving cell classification."

Supplementary Figure 10. Additional representative cell classification results based on SSCOR-corrected images. **a** The complete correction results of the real H&E image in Fig. 5c, and the enlarged classification results of another stripe-free region (white box). Different colours of the nuclear boundaries denote separate instances. The white arrows point to the classified tumor cells, as confirmed by experienced pathologists. Scale bar: 1mm. **b** The correction and classification results of a synthetic stripe from the H&E dataset, CoNSep. The green arrows indicate that the cell classification of SSCOR is consistent with GT. **c** Four quantitative metrics further demonstrated that SSCOR-corrected images have the capability of improving segmentation and classification performance. F_c^i and F_c^s denote the F_1 classification score for the inflammatory and spindle-shaped nuclei classes respectively. Dice coefficient (DICE) measures the separation of nuclei from the background. Panoptic quality (PQ) represents a unified score for measuring the performance of nuclear instance segmentation methods.

REVIEWERS' COMMENTS

Reviewer #1 (Remarks to the Author):

The authors has conducted beneficial explorations in the field of strip correction. Thank for their meticulous and patient response.

However, it is very difficult to realize automatic correction of microscopic image stripes so far, because it involves different imaging conditions and equipment as well as parameters selection and amplification resolution, etc. Second, the handling of complex situations should be further studied. The selection of normal image blocks and image quality evaluation are very important and should be further studied deeply.

The estimation of the position of the strip is not a simple matter, especially in the presence of sub-normal stripes, mixed with multiple stripes and artificial effects. The above issues require further in-depth research.

Reviewer #2 (Remarks to the Author):

The authors have address all of my prior concerns. The manuscript is now acceptable for publication.

Reviewer #3 (Remarks to the Author):

I think the authors' effort to address the comments. Most of the comments are addressed in the rebuttal. While as also pointed out by R1, this framework is not an automatic method, due to the human intervention and guidance to choose normal/abnormal patches for different stripe types and different sampling strategies.

I acknowledge it is difficult to develop such a fully automatic framework. But for a better position of this paper, the authors should list these points as explicit weaknesses and explicitly discuss them in the manuscript. Moreover, some other response materials can also be included in the manuscript due to the valuable information, such as the figures to demonstrate the training procedure.

Reply Letter

We appreciate the time and efforts from the reviewers. We have carefully considered all the comments and suggestions raised by the reviewers and have addressed them point-by-point. The corresponding changes to the resubmitted main manuscript have been highlighted in red.

Reviewer #1

The authors have conducted beneficial explorations in the field of strip correction. Thank for their meticulous and patient response.

However, it is very difficult to realize automatic correction of microscopic image stripes so far, because it involves different imaging conditions and equipment as well as parameters selection and amplification resolution, etc. Second, the handling of complex situations should be further studied. The selection of normal image blocks and image quality evaluation are very important and should be further studied deeply.

The estimation of the position of the strip is not a simple matter, especially in the presence of sub-normal stripes, mixed with multiple stripes and artificial effects. The above issues require further in-depth research.

Response: we appreciate the valuable feedback from the reviewer, which has undoubtedly helped to improve the quality of our manuscript. Firstly, regarding the difficulty of achieving automatic correction, we have elucidated this limitation in the Discussion section. Secondly, for investigating the performance on complex situations, we have conducted comprehensive experiments to demonstrate the effectiveness of our model in handling non-uniform/grid/oblique stripes and out-of-focus/bubble-like/SFA artifacts, as well as the complex ones with stripes and artifacts combined (Supplementary Fig. 7). Please refer to the updated Discussion section as shown below,

"Nevertheless, there still remains several limitations in the proposed approach. **First, SSCOR offers a semi-automated approach rather than a fully automated one, which necessitates a minor degree of human intervention during the initial phases of training. This intervention is primarily required to approximately determine the positions of stripes and the appropriate patch size. As the future work, we aim to make SSCOR fully automatic while maintaining its generalizability and effectiveness.**"

As for the normal patch selection, we conducted experiments to study the effect of patch size (Proximity Sampling in Supplementary Note 4). In the revised version, we have included the analysis on the step size of sampling patches in Supplementary Information, as shown in Supplementary Fig. 16.

Supplementary Figure 16. The effect of sampling step size on model performance. a) and b) For comparison, we applied different step sizes during training, and SSCOR showed the stable performance with the step sizes of 64, 128, or 256. Despite a slight performance drop at the step size of 512, SSCOR is able to well correct stripes. Note that, the white arrow indicates the partially uncorrected stripes when the step size is set as 512. Scale bar of a): 1 mm. Scale bar of b): 500 μm . c) The PSNR and SSIM curves for the corrected images exhibit little fluctuation, indicating the stability of SSCOR.

Regarding the image quality evaluation, existing non-reference image quality assessment methods can hardly reflect the correction quality of stitched microscope stripe images accurately. In addition to the adopted metrics like ICV, we have included a comprehensive user study for image quality assessment (Supplementary Note 3. User study). As the future work, we will further study a specialized assessment method.

Lastly, we acknowledge that estimating stripe positions can sometimes be challenging. In the "high tolerance towards imprecise prior stitched information" subsection in Results, we have clarified that our model is designed to be tolerant towards mild estimation errors. Besides, to enhance the utilization of SSCOR, we have emphasized the significance of normal patches and stripe position. Please refer to Discussion section as below,

"In practice, the utilization of SSCOR involves two fundamental steps: (1) rough estimation of stripe positions and (2) selection of normal patches. To optimize the usability and efficiency of our method, we have outlined several guidelines that encompass crucial elements for successful implementation in the following aspects: a schematic diagram of typical stripes and artifacts (Supplementary Fig. 13), the representative sampling cases of stripe and artifact (Supplementary Fig. 1), as well as the corresponding description and sampling strategy (Supplementary Table 4). The provided guidelines empower researchers to effectively sample patches for different stripes and artifacts with minimal manual intervention."

Reviewer #2

The authors have address all of my prior concerns. The manuscript is now acceptable for publication.

Response: Thank you for all the review comments and advice.

Reviewer #3

I think the authors' effort to address the comments. Most of the comments are addressed in the rebuttal. While as also pointed out by R1, this framework is not an automatic method, due to the human intervention and guidance to choose normal/abnormal patches for different stripe types and different sampling strategies.

I acknowledge it is difficult to develop such a fully automatic framework. But for a better position of this paper, the authors should list these points as explicit weaknesses and explicitly discuss them in the manuscript. Moreover, some other response materials can also be included in the manuscript due to the valuable information, such as the figures to demonstrate the training procedure.

Response: I would like to express my sincere gratitude for your positive feedback regarding our

manuscript. In the revised version, we have provided explicit clarification and discussion related to the limitation that SSCOR cannot achieve fully automated correction in the Discussion section. In addition, as suggested by reviewer, we have included a detailed explanation of the training procedures (Supplementary Fig. 15), along with the analysis of the step size of sampling (Supplementary Fig. 16), in Supplementary Information, respectively. We hope that these additions further enhance the clarity and comprehensiveness of our work. Please see Network Implementation Details in the Methods section.

"Note that, we did not utilize any pre-trained weights during the training process. The training procedure is stable even with different random initializations (Supplementary Fig. 15). ... The optimal step size for patch sampling is determined according to the experimental results in Supplementary Fig. 16."

Supplementary Figure 15. Model performance versus the training epochs with random initialization. We show the training curves with regards to the correction quality for the synthesized stripes and out-of-focus artifacts (Fig. 4a), respectively. For each synthesized image, model training is performed for three times with random initialization, which was measured in PSNR and SSIM for every 10 epochs until it reaches 200 epochs. As observed, both PSNR and SSIM exhibit an increasing trend with more training epochs. Although three training curves show fluctuations during training process, they eventually converge to similar results at the 200th epoch.